# Constrained Feedback Learning for Non-Stationary Multi-Armed Bandits

**Shaoang Li**
Stony Brook University
shaoang.li@stonybrook.edu

**Jian Li**
Stony Brook University
jian.li.3@stonybrook.edu

## Abstract

Non-stationary multi-armed bandits (NSMAB) enable agents to adapt to changing environments by incorporating mechanisms to detect and respond to shifts in reward distributions, making them well-suited for dynamic settings. However, existing approaches typically assume that reward feedback is available at every round—an assumption that overlooks many real-world scenarios where feedback is limited. In this paper, we take a significant step forward by introducing a new model of *constrained feedback in non-stationary multi-armed bandits* (CONFEE-NSMAB), where the availability of reward feedback is restricted. We propose the first prior-free algorithm—that is, one that does not require prior knowledge of the degree of non-stationarity—that achieves near-optimal dynamic regret in this setting. Specifically, our algorithm attains a dynamic regret of $\tilde{\mathcal{O}}(K^{1/3}V_T^{1/3}T/B^{1/3})$, where $T$ is the number of rounds, $K$ is the number of arms, $B$ is the query budget, and $V_T$ is the variation budget capturing the degree of non-stationarity.

## 1   Introduction

The multi-armed bandits (MAB) problem [29] is a fundamental framework for decision-making under uncertainty, where an agent selects from a set of arms to maximize cumulative rewards over a time horizon, balancing exploration (learning about arms) and exploitation (leveraging known rewards). Most state-of-the-art MAB algorithms, including Upper Confidence Bound (UCB)-type methods [1] and Thompson Sampling (TS)-based approaches [34], assume stationary settings in which reward distributions remain fixed over time. However, many real-world applications—such as dynamic pricing [14], evolving user preferences [21], and environmental shifts [16]—are inherently non-stationary, with reward distributions that change over time. This has led to growing interest in non-stationary MAB (NSMAB), which incorporates mechanisms for detecting and adapting to such changes, making it more applicable to dynamic environments.

However, most of the existing literature on NSMAB assumes that reward feedback is available to the agent at every round. This overlooks a critical aspect of many real-world applications, where feedback is often limited. For example, in recommendation systems, repeatedly asking users for feedback on the quality of recommendations can lead to user fatigue or annoyance [13]. Similarly, in reinforcement learning from human feedback (RLHF) [11], essential signals such as preference comparisons, demonstrations, or ratings are often costly, time-consuming, or constrained by the availability of expert annotators. Motivated by these challenges, we introduce a new model of *constrained feedback in non-stationary multi-armed bandits*, termed CONFEE-NSMAB, in which the availability of reward feedback is limited.

Similar to existing NSMAB frameworks, CONFEE-NSMAB quantifies the degree of non-stationarity in the environment by tracking changes in reward distributions and assumes that the variation in mean rewards over the time horizon $T$ is bounded by a variation budget $V_T$ [5]. However, unlike conventional NSMAB, the agent in CONFEE-NSMAB cannot observe reward feedback at every round.

39th Conference on Neural Information Processing Systems (NeurIPS 2025).

| Paper | Setting | Variation $V_T$ | Constrained Feedback | Upper Bound | Lower Bound |
|---|---|---|---|---|---|
| Besbes et al. [5] | Non-Stationary | Known | ✗ | $\tilde{\mathcal{O}}(K^{1/3}V_T^{1/3}T^{2/3})$ | $\Omega(K^{1/3}V_T^{1/3}T^{2/3})$ |
| Wei and Luo [37] | Non-Stationary | Unknown | ✗ | $\tilde{\mathcal{O}}(K^{1/3}V_T^{1/3}T^{2/3})$ | $\Omega(K^{1/3}V_T^{1/3}T^{2/3})$ |
| Efroni et al. [13] | Stationary | — | ✓ | $\tilde{\mathcal{O}}\left(\frac{K^{1/2}T}{B^{1/2}}\right)$ | $\Omega\left(\frac{K^{1/2}T}{B^{1/2}}\right)$ |
| **This Work** | **Non-Stationary** | **Unknown** | ✓ | $\tilde{\mathcal{O}}\left(\frac{K^{1/3}V_T^{1/3}T}{B^{1/3}}\right)$ | $\Omega\left(\frac{K^{1/3}V_T^{1/3}T}{B^{1/3}}\right)$ |

Table 1: Comparison with the most closely related works, where $T$ is the time horizon, $K$ is the number of arms, $V_T$ denotes the variation budget capturing the degree of non-stationarity, and $B$ is the reward feedback querying constraint.

Instead, *the total number of reward feedback queries available to the agent over the time horizon $T$ is limited by a reward query budget $B$*. This constraint provides the agent with significantly less information than in standard NSMAB settings, making the CONFEE-NSMAB problem fundamentally more challenging. Motivated by this, the primary question we seek to address is:

> *How does the reward feedback querying constraint affect the agent's decision-making process in* CONFEE-NSMAB*, particularly in balancing exploration, exploitation, and query efficiency under limited feedback?*

Despite recent advances in feedback-constrained MAB [13, 26] and prior-free[1] non-stationary RL or MAB [37], this question remains, to the best of our knowledge, unexplored in the context of constrained feedback learning for NSMAB (see Table 1). In this setting, the agent faces a unique dilemma: balancing exploration and exploitation in non-stationary environments while operating under a strict feedback query constraint. This introduces a new layer of complexity to the already challenging problem of designing prior-free algorithms for NSMAB with near-optimal dynamic regret[2] guarantees. This leads us to the second research question:

> *Is it possible to design prior-free algorithms for* CONFEE-NSMAB *that achieve near-optimal dynamic regret guarantees?*

Motivated by these open questions, we make the following key contributions:

• **CONFEE-NSMAB Framework.** We introduce CONFEE-NSMAB, a new framework for NSMAB that incorporates a reward feedback querying constraint, limiting the total number of reward feedback queries the agent can issue over the time horizon while learning in non-stationary environments.

• **Near-Optimal HYQUE Algorithm.** A standard approach in NSMAB is to run multiple instances of a base algorithm at varying time scales using randomized scheduling to detect non-stationarity [37]. We extend this multi-scale idea to CONFEE-NSMAB, but the extension presents two key challenges:

▷ Query Allocation Trade-off: Naively querying every round quickly exhausts the budget, while querying too infrequently risks missing abrupt changes. Thus, the challenge is to design a query allocation strategy that balances timely detection with budget preservation, including careful distribution across time scales.

▷ Long Non-Query Segments: Longer instances aimed at capturing gradual shifts may contain extended non-query periods, limiting the algorithm's responsiveness. It is critical to ensure sufficient query frequency for timely change detection without prematurely exhausting the budget.

To address these challenges, we propose HYQUE, a hybrid query allocation algorithm that combines baseline allocation—ensuring a minimum query rate within each block—with an on-demand mechanism that dynamically injects queries when usage falls behind a near-linear pace. HYQUE is prior-free and provably near-optimal, achieving a regret upper bound that matches the lower bound up to logarithmic factors. We establish this lower bound to understand how the reward feedback querying constraint impacts the best-achievable dynamic regret in CONFEE-NSMAB.

---

[1]The algorithm operates without prior knowledge of the degree of non-stationarity, making it more practical and broadly applicable.

[2]In non-stationary settings, the conventional "static regret", defined with respect to a single best action, may perform poorly. Instead, "dynamic regret" is the appropriate performance metric [5]. See Section 2 for a detailed definition.

## 2 Model and Problem Formulation

**NSMAB.** We consider a NSMAB setting with $[K] = \{1, \ldots, K\}$ arms, where an agent selects one arm $k \in [K]$ at each round $t \in [T] = \{1, \ldots, T\}$. When arm $k$ is pulled, a reward $R_t^k \in [0, 1]$ is obtained, where $R_t^k$ is a random variable drawn independently from an *unknown* distribution. Let $\mu_t^k$ denote the expected reward of arm $k$ at round $t$. Let $\mu := \{\mu_t^k, k \in [K], t \in [T]\}$ denote the underlying sequence of true expected rewards for all arms over the time horizon $T$. In NSMAB, $\mu_t^k$ may vary across rounds. Consistent with conventional NSMAB settings [5], while the expected rewards of each arm may change arbitrarily often, the total variation in the expected rewards over $[T]$ is bounded by a variation budget $V_T$, whose value is unknown to the agent. The corresponding non-stationarity set can be defined as $\mathcal{V} = \left\{ \mu : \sum_{t=1}^{T-1} \sup_{k \in [K]} \left| \mu_{t+1}^k - \mu_t^k \right| \leq V_T \right\}$.

**NSMAB with Constrained Feedback (CONFEE-NSMAB).** At each round $t$, the agent performs two actions: selects an arm $k \in [K]$ to pull, and decides whether to query the reward feedback subject to the budget constraint. Specifically, we assume that querying reward feedback incurs a unit cost, and define $\mathbb{I}_t^{\text{query}} \in \{0, 1\}$ as an indicator variable denoting whether the reward feedback is queried at round $t$. The agent can observe the reward $R_t^k$ immediately only if $\mathbb{I}_t^{\text{query}} = 1$. The total number of queries over the time horizon $T$ is constrained by a known query budget $B$, i.e., $\sum_{t=1}^{T} \mathbb{I}_t^{\text{query}} \leq B$. An algorithm $\mathcal{A}$ is considered feasible if it selects an arm $k \in [K]$ at each round $t$ and adheres to the total query budget constraint. Throughout the paper, we use $\mathcal{S}^{\text{query}} = \{t \in [T] \mid \mathbb{I}_t^{\text{query}} = 1\}$ and $\mathcal{S}^{\text{non-query}} = \{t \in [T] \mid \mathbb{I}_t^{\text{query}} = 0\}$ to denote the sets of query and non-query rounds, respectively.

**Dynamic Regret.** Let $\mu_t^* = \max_{k \in [K]} \mu_t^k$ denote the optimal expected reward at round $t$. The performance of the agent is evaluated using the *dynamic regret*. This metric compares the cumulative expected reward of an optimal policy (aware of $\mu_t^*$ at each round) with that accrued from the sequence of arms selected by algorithm $\mathcal{A}$. Formally, it is defined as

$$\mathcal{R}_T(\mathcal{A}) = \sup_{\mu \in \mathcal{V}} \left\{ \sum_{t=1}^{T} \mu_t^* - \mathbb{E}\left[ \sum_{t=1}^{T} \mu_t^k \right] \right\}, \tag{1}$$

where the expectation is taken over the randomness in rewards and potentially the internal randomness of the algorithm. The objective of the agent in CONFEE-NSMAB is to minimize the dynamic regret. While this definition is similar to those in existing NSMAB frameworks, the key distinction is that, in CONFEE-NSMAB, the agent cannot always observe the reward feedback at each round. Instead, querying the reward feedback incurs a cost, which is constrained by a budget $B$ over the time horizon $T$ (i.e., $\sum_{t=1}^{T} \mathbb{I}_t^{\text{query}} \leq B$.). These fundamental differences necessitate new techniques for online algorithm design and dynamic regret characterization, which we will discuss in detail in Sections 3 and 4.1, respectively.

## 3 Prior-free Algorithm for CONFEE-NSMAB

We show that it is possible to design a prior-free algorithm for CONFEE-NSMAB that achieves near-optimal guarantees on dynamic regret.

### 3.1 Motivation and Challenges

We begin by outlining the core principles that guide our approach. The primary objective is to *strike a balance between accurately detecting environmental changes and efficiently managing a limited query budget.*

● **Balancing Change Detection and Query Usage.** In NSMAB, the algorithm must detect environmental changes by tracking shifts in the mean rewards of the arms. However, when the total number of queries is constrained, naive strategies such as querying every round quickly exhaust the query budget. Conversely, querying too infrequently may hinder the algorithm's ability to detect abrupt changes in a timely manner. Effectively allocating queries across multiple time scales without exceeding the budget $B$ requires a carefully crafted strategy. ***The central challenge, therefore, is to develop a query allocation mechanism that balances the trade-off between aggressive exploration (to detect changes) and conservative query usage (to preserve the budget).***

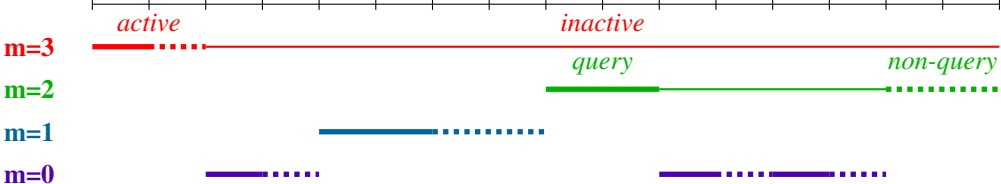

Figure 1: An illustration of a single block in BAQUE with $n = 3$ and $b = 2$ is shown. The figure spans a total of 16 rounds ($2^3 \cdot 2$) and highlights the hierarchical structure of the procedure across four scales ($m = 0, 1, 2, 3$), where each scale alternates between active and inactive states. The illustration includes 1 instance at $m = 3$, 1 instance at $m = 2$, 1 instance at $m = 1$, and 3 instances at $m = 0$. Bold solid lines indicate active periods, while thin solid lines represent inactive periods. Active periods are further divided into two segments—continuous and dotted sections—to represent distinct querying and non-querying rounds.

- **Budget Splitting across Multiple Scales.** Conventional prior-free NSMAB approaches often employ a *multi-scale restart* technique, where multiple instances of a base algorithm (e.g., UCB1 [1]) are run at exponentially increasing time scales [10, 37]. This ensures that at least one instance operates at an appropriate granularity to detect changes. ***As a result, it becomes essential to allocate the query budget across different scales and their corresponding base algorithms***. This allocation serves a dual purpose: it ensures that the regret incurred during query segments remains near-optimal—comparable to settings with full feedback—and it supports the accuracy of decisions made in non-query segments, which rely entirely on the reward estimates obtained during query phases.

- **Balancing Query Frequency.** For longer instances designed to capture subtle environmental changes, there may be extended non-query segments. When the agent encounters a prolonged sequence of non-query rounds, it may be unable to verify whether the environment has shifted, leading to potentially significant regret if the chosen arm becomes suboptimal due to an undetected change. Consequently, non-query gaps cannot be arbitrarily long without degrading performance. The algorithm must query frequently enough to detect changes in a timely manner, while ensuring that the query frequency is not so high as to prematurely exhaust the total query budget $B$.

### 3.2 HYQUE Algorithm

**Overview.** To address these challenges, we propose HYQUE, a *hybrid query allocation* algorithm for CONFEE-NSMAB. HYQUE partitions the time horizon into blocks of geometrically increasing lengths, within which multiple parallel instances of a base algorithm (e.g., UCB1) operate at different temporal scales. Small-scale instances are more responsive to rapid changes and emphasize exploration, while large-scale instances target slower trends and focus on exploitation. Since the block length upper-bounds the instance duration, this structure enables systematic query scheduling that balances adaptability to non-stationarity, effective exploration, and budget conservation.

At the block level, HYQUE invokes a *baseline query allocation* subroutine (BAQUE) (Section 3.2.1), which guarantees a minimal share of queries across scales to ensure robust change detection. On top of this, HYQUE incorporates an *on-demand query allocation* mechanism (Section 3.2.2) that continuously tracks cumulative query usage and injects additional queries when usage falls behind a near-linear pacing schedule. This two-tiered design offers worst-case robustness within each block while adaptively adjusting to smoother environments, achieving a principled trade-off between stability and flexibility in query allocation. We begin by detailing the BAQUE subroutine.

#### 3.2.1 The BAQUE Subroutine

To ensure that each time-scale instance receives sufficient queries for detecting abrupt changes, we *pre-allocate* a baseline portion of the query budget. Instead of distributing queries across the entire time horizon, allocation is performed at the block level, covering instances operating at different scales. We guarantee that within the active rounds of each instance, approximately a $\frac{B}{2T}$ fraction is designated for query rounds.

Although this proportion may appear small, it ensures that no instance is deprived of queries, thereby preserving HYQUE 's ability to adapt to frequent or significant environmental changes. Typically,

each instance begins with an initial *one-shot query batch*, in which the algorithm uses its baseline query quota to collect updated reward observations (Line 10 of Algorithm 1). This initial query batch is crucial for enabling reliable change detection at the corresponding scale.

By distributing queries in this manner, each scale is guaranteed a sufficient number of queries over the time horizon, ensuring its ability to detect significant changes (Lemma 4.6). Moreover, for long-duration instances with extended non-query periods, the probabilistic creation of shorter-scale instances—each beginning with its own query batch—naturally segments the timeline into overlapping sub-intervals. These overlaps help prevent prolonged periods without feedback, as such uninterrupted non-query spans are unlikely to persist with high probability (Lemma 4.4).

---

**Algorithm 1** BAQUE: Baseline Query Allocation

---

**Require:** Query budget ratio $b = \lceil 2T/B \rceil$, time-scale parameter $n$.
1: **for** $\tau = 0, b - 1, 2b - 1, \ldots, 2^n \cdot b - 1$ **do**
2:      **for** $m = n, n - 1, \ldots, 0$ **do**
3:          **if** $\tau$ is a multiple of $b \cdot 2^m$ **then**
4:              With probability $2^{\frac{m-n}{2}}$, initiate a new instance $\mathcal{I}_{n,m,\tau}$ spanning rounds $[\tau + 1, \tau + b \cdot 2^m]$;
5: **for** each instance $\mathcal{I}_{n,m,\tau}$ **do**
6:      Let $\mathcal{S}_{n,m,\tau}$ be its *active* rounds;
7:      **Query batch**: For the first $\max(1, \lfloor |\mathcal{S}_{n,m,\tau}|/b \rfloor)$ active rounds, run UCB1 (collect reward and update the UCB index, denoted as $\tilde{g}_t$);
8:      **Non-query batch**: In the remaining active rounds, pick arms according to their frequencies from query batch (no reward feedback).

---

Algorithm 1 constructs a *block* of length $b \cdot 2^n$ rounds and deploys multiple instances of UCB1 across various time scales indexed by $m = 0, 1, \ldots, n$. As illustrated in Fig. 1, these instances operate at different granularities, resulting in overlapping activity that alternates between query and non-query rounds. The multi-scale instance setup (Lines 1–7) proceeds by iterating through the timeline in steps of $b$, and at each time $\tau$, a new instance $\mathcal{I}_{n,m,\tau}$ is probabilistically initiated if $\tau$ is divisible by $b \cdot 2^m$. This randomized activation ensures coverage across both short- and long-term scales, allowing the algorithm to adapt to diverse patterns of non-stationarity. Once initiated (Line 4), an instance $\mathcal{I}_{n,m,\tau}$ covers the nominal time span $[\tau + 1, \tau + b \cdot 2^m]$. The set of rounds where the instance is truly *active*, denoted $\mathcal{S}_{n,m,\tau}$, is determined through a hierarchical masking mechanism: a round $t$ within the nominal span is included in $\mathcal{S}_{n,m,\tau}$ if and only if it is not covered by any initiated instance operating at a finer scale (i.e., with $m' < m$). This ensures that finer-scale instances take precedence over coarser ones in shared intervals.

The baseline query allocation phase (Lines 8–12) then operates on these active rounds. For each instance $\mathcal{I}_{n,m,\tau}$, its (potentially non-contiguous) active rounds $\mathcal{S}_{n,m,\tau}$ are split into two parts (Lines 10–12). The first is a *query batch* of size $\max(1, \lfloor |\mathcal{S}_{n,m,\tau}|/b \rfloor)$, during which UCB1 is actively invoked and reward feedback is collected. The second is a *non-query batch*, comprising the remaining rounds, where actions are selected according to the empirical frequency distribution of arms from the query batch. For example, if a particular arm was selected in 20 out of 50 query rounds, it will be selected in the non-query phase with probability 0.4—without incurring additional feedback cost. Our choice of UCB1 as the base algorithm and frequency-based sampling for the non-query phase is motivated by the goal of providing a clear theoretical analysis. In practice, these components are modular and can be replaced by other choices, such as a different base algorithm or an alternative sampling strategy like a greedy policy that exclusively selects the most frequently chosen arm from the query phase.

### 3.2.2 The HYQUE Framework

Although BAQUE ensures robust performance in worst-case scenarios, it can be overly conservative when changes in the environment are infrequent or relatively mild. To better leverage such settings, we introduce HYQUE (Algorithm 2), which augments BAQUE with an *on-demand query allocation mechanism*.

Intuitively, when the on-demand component detects that the cumulative number of queries used so far falls short of a target proportion—specifically, $\frac{t}{T} \cdot B$ at time $t$—it allocates additional query rounds

at the current scale to refine reward estimates. This mechanism promotes *near-linear pacing* of query usage over time, ensuring that total query consumption closely tracks the ideal rate of $\frac{B}{T}$, thus avoiding both excessive querying and underutilization. To account for stochastic variability, a buffer term $\min\{T/\sqrt{B}, 2^n, T - t\}$ is subtracted from the threshold. This guards against over-querying in short segments and prevents abrupt spikes in query activity (see Remark 4.5 and Lemma 4.7). As a result, HYQUE is able to opportunistically allocate more queries during stable periods, enabling finer reward estimation and lower regret in environments that do not change drastically.

---

**Algorithm 2** HYQUE: Hybrid Query Allocation

---
1: **Initialize:** current round $t \leftarrow 1$, used queries $B' \leftarrow 0$.
2: **for** $n = 0, 1, \ldots$ **do**
3:     Set $t_n \leftarrow t$ and initialize BAQUE for block $[t_n, t_n + b \cdot 2^n - 1]$ with the time-scale parameter $n$;
4:     **while** $t < t_n + b \cdot 2^n$ **do**
5:         **if** current instance has queries **then**
6:             Receive the UCB index $\tilde{g}_t$ and the selected arm from BAQUE, play the arm, then increment $t$ and $B'$;
7:             Update the BAQUE instance with the observed feedback;
8:             Perform environmental change tests. If any test fails, restart a new phase from *Line 2*;
9:         **else**
10:             **if** $B' < \frac{t \cdot B}{T} - \min\{T/\sqrt{B}, 2^n, T - t\}$ **then**
11:                 Convert the current non-query round into a query round and jump to *Line 6*;
12:             No feedback requested; and set $t \leftarrow t + 1$.

---

HYQUE partitions the time horizon into multiple phases, each initiated upon detecting a change in the environment. Within each phase, time is further divided into consecutive blocks of length $b \cdot 2^n$ rounds. At the beginning of each block, HYQUE invokes BAQUE to initialize multi-scale instances and records the block's start time as $t_n$, so the block spans the interval $[t_n, t_n + b \cdot 2^n - 1]$.

During each block, HYQUE actively monitors two key aspects to determine whether to restart the phase or allocate additional queries.

(1) Change Detection (Lines 5–9 of Algorithm 2): For a given instance $\mathcal{I}_{n,m,\tau}$, define $\mathcal{S}_{n,m,t} = \{t' \mid t_n \leq t' \leq t; t' \in \mathcal{S}^{\text{query}}\}$ as the set of its query rounds up to time $t$. Let $U_t = \min_{t' \in \mathcal{S}_{n,m,t}} \tilde{g}_{t'}$ be the minimum estimated reward, and define the confidence bound $\hat{\rho}(t) = 6(\log T + 1) \log\left(\frac{T}{\delta}\right) \left( \sqrt{\frac{K \log t}{t}} + \frac{K}{t} \right)$. Let start and end denote the first and last query rounds of the instance, respectively. Two tests are performed: (i) If $t = $ end for some order-$m$ instance and $\frac{1}{2^m} \sum_{t' \in \mathcal{S}_{n,m,\tau}} R_{t'} \geq U_t + 9\hat{\rho}(2^m)$, the test returns fail; and (ii) If the average discrepancy satisfies $\frac{1}{|\mathcal{S}_{n,m,t}|} \sum_{t' \in \mathcal{S}_{n,m,t}} (\tilde{g}_{t'} - R_{t'}) \geq 3\hat{\rho}(|\mathcal{S}_{n,m,t}|)$, the test also returns fail.

(2) On-Demand Query Activation (Lines 10–12): Let $B'$ be the total number of queries used up to round $t$. If $B'$ falls significantly below the expected usage $\frac{t \cdot B}{T}$—adjusted by a buffer term $\min\left\{ \frac{T}{\sqrt{B}}, 2^n, T - t \right\}$—then HYQUE allocates an additional query to the active instance. This buffer accounts for: (i) $\frac{T}{\sqrt{B}}$: the maximum consecutive query rounds needed to ensure worst-case performance (Remark 4.5); (ii) $2^n$: the maximum size of any single query batch in the current block; and (iii) $T - t$: the remaining number of rounds, ensuring budget feasibility.

## 4 Main Theoretical Result

In this section, we bound the regret of the HYQUE Algorithm.

**Theorem 4.1.** *For* CONFEE-NSMAB *with a query budget $B = \omega(T^{3/4})$ and an unknown variation satisfying $V_T \in [K^{-1}, K^{-1}\sqrt{B}]$, our* HYQUE *utilizes at most $B$ queries, and its regret is bounded with high probability as follows:*

$$\mathcal{R}_T(\text{HYQUE}) \leq \tilde{\mathcal{O}}\left( \frac{K^{1/3} V_T^{1/3} T}{B^{1/3}} \right). \tag{2}$$

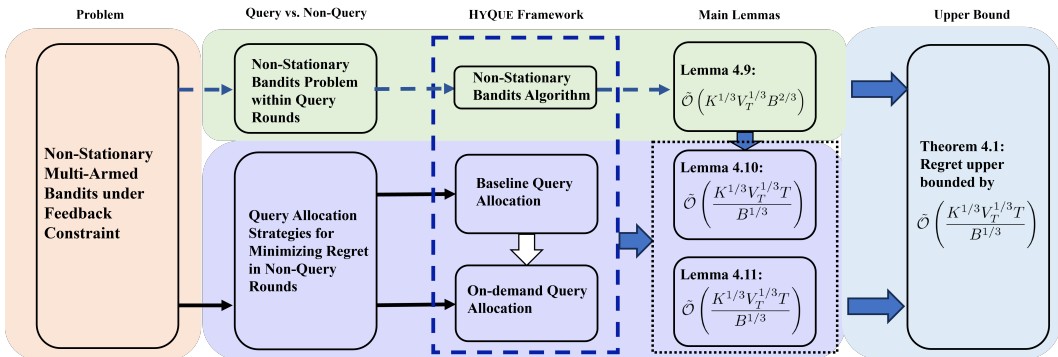

Figure 2: The workflow of HYQUE for CONFEE-NSMAB and its corresponding regret analysis. The top (green) box addresses the NSMAB problem during query rounds, which leads to the regret bound established in Lemma 4.9. The bottom (purple) box describes the query allocation strategies within HYQUE, where BAQUE operates as a subroutine. Together, they aim to minimize regret during non-query rounds, as analyzed in Lemmas 4.10 and 4.11. By combining these results, Theorem 4.1 provides the overall regret bound for our HYQUE algorithm.

Ignoring logarithmic factors, this upper matches the lower bound in Theorem 5.1 in terms of its dependence on $K$, $V_T$, $T$, and $B$, thereby establishing the near-optimality of HYQUE. When $B = \Omega(T)$, the regret upper bound simplifies to $\tilde{O}\left(K^{1/3}V_T^{1/3}T^{2/3}\right)$, recovering the best-known result for NSMAB [5].

As discussed earlier (Section 3.1), our HYQUE must balance environmental change detection with query allocation due to the presence of the query constraint in CONFEE-NSMAB. This key distinction renders existing regret analysis for NSMAB [5, 37] not directly applicable, and necessitates the development of new proof techniques tailored to constrained feedback settings. First, establishing feasibility (Section 4.1.1) requires a careful analysis of query usage, made more challenging by the inherent uncertainty in the non-stationary environment. Second, we introduce a novel regret decomposition (Section 4.1.2) that enables us to isolate and address the unique structural challenges posed by limited feedback, allowing separate treatment of regret incurred during query and non-query rounds. Third, building upon this decomposition, we develop fine-grained analytical techniques (Section 4.1.3) to rigorously bound the regret contributed by non-query rounds—an essential step in accurately characterizing performance under constrained feedback. An overview of our theoretical analysis is presented in Section 4.1, with the complete proof detailed in Appendix A.

A lower bound on the budget $B$ is necessary because non-stationary bandit problems can exhibit a mixture of both abrupt and gradual changes. Consequently, any algorithm aiming for optimal regret must detect all such changes with high probability, which in turn requires a sufficiently large budget $B$. Specifically for the HYQUE algorithm, a larger $B$ ensures that its randomized querying strategy provides adequate coverage across the entire time horizon. Furthermore, it facilitates a cleaner analysis, allowing us to derive an explicit expression for the algorithm's success probability and to guarantee a strong high-probability bound, such as $\Pr(\text{optimal regret}) \geq 1 - \exp(-T^{\Omega(1)})$.

*Remark* 4.2. If $V_T$ is *known* in CONFEE-NSMAB, a near-optimal upper bound can similarly be achieved, and the algorithm becomes significantly simpler. A detailed description of this algorithm, along with a rigorous analysis of its theoretical regret bound, is provided in Appendix B.

*Remark* 4.3. When $B = T$, allowing queries at every round, HYQUE's behavior on query rounds aligns with the principles of the MASTER algorithm [37], the only known prior-free solution for non-stationary RL. As a result, the query allocation component of HYQUE primarily affects the regret incurred during the non-query segments.

## 4.1 Proof Sketch

We provide an overview of the theoretical analysis for the regret upper bound for HYQUE.

### 4.1.1 Feasibility of HYQUE

**Lemma 4.4** (Bound on Consecutive Non-query Rounds). *For some universal constant $C > 0$, the probability that BAQUE experiences more than $T/\sqrt{B}$ consecutive non-query rounds in that block is at most $\exp\left(-C\frac{B^{3/4}}{\sqrt{T}}\right)$.*

Instances initiated by BAQUE start with a query phase. The probability of a specific potential instance $\mathcal{I}_{n,m,\tau}$ (for block parameter $n$, scale $m$) *not* being initiated is $1 - 2^{\frac{m-n}{2}}$. Consequently, by bounding the probability that no new instance is initiated over a period of $T/\sqrt{B}$ rounds, we establish an upper bound on the probability that this period consists entirely of non-query rounds (as any new instance would have introduced queries).

*Remark* 4.5. The same analysis applies to query rounds as well. In particular, with $B = o(T)$, the probability that BAQUE undergoes more than $T/\sqrt{B}$ consecutive *query* rounds is also at most $\exp\left(-C\frac{B^{3/4}}{\sqrt{T}}\right)$.

**Lemma 4.6** (Stability in Each Phase). *Consider any phase of HYQUE. Throughout this phase, the fraction of query rounds allocated by BAQUE, relative to the total number of rounds in the same phase, satisfies: $\frac{1}{2} \cdot \frac{B}{T} \leq \frac{\text{(\# of query rounds in the phase)}}{\text{(\# of total rounds in the phase)}} < \frac{B}{T}$.*

**Lemma 4.7** (Query Budget Feasibility). *For HYQUE, the total number of query rounds throughout the time horizon $T$ does not exceed $B$.*

In each phase, HYQUE maintains the query allocation within the proportion $B/T$ for every block except potentially the last one, where additional caution is needed to handle boundary effects. In the final block, to avoid a large run of consecutive query rounds following an immediate restart, HYQUE employs a "buffer term" as discussed in Section 3.2.2. This design ensures that the query allocation does not exceed the proportion $B/T$. Consequently, by combining BAQUE with on-demand allocation, HYQUE ensures that the total number of queries used does not exceed $B$ over the entire time horizon $T$. A detailed proof is provided in Appendix A.1.

### 4.1.2 Regret Decomposition

We decompose the dynamic regret $\mathcal{R}_T$ into three parts:

**Lemma 4.8.** *For any algorithm $\mathcal{A}$, we define $\mathcal{S}_{t-1} \subseteq \mathcal{S}^{query} \cap \{1, \ldots, t-1\}$ as the set of query rounds whose observed feedback is used by the algorithm at round $t$. Consequently, the arm selection $a_t$ is a function $f_t(\mathcal{H}_{t-1})$ of the history $\mathcal{H}_{t-1} = \{(a_s, r_s) : s \in \mathcal{S}_{t-1}\}$. The regret in (1) can then be upper bounded by:*

$$\mathcal{R}_T \leq \underbrace{\sum_{t \in \mathcal{S}^{query}} \mu_t^* - \mathbb{E}\left[\sum_{t \in \mathcal{S}^{query}} R_t\right]}_{\mathcal{R}_T^{query}} + \underbrace{\sum_{t \in \mathcal{S}^{non\text{-}query}} \mathbb{E}\left[\max_{k \in [K]} \sum_{t \in \mathcal{S}_{t-1}} \frac{R_t^k}{|\mathcal{S}_{t-1}|}\right] - \mathbb{E}\left[\sum_{t \in \mathcal{S}^{non\text{-}query}} R_t\right]}_{\mathcal{R}_T^{error}}$$

$$+ \underbrace{\sum_{t \in \mathcal{S}^{non\text{-}query}} \mu_t^* - \sum_{t \in \mathcal{S}^{non\text{-}query}} \mathbb{E}\left[\max_{k \in [K]} \sum_{t \in \mathcal{S}_{t-1}} \frac{R_t^k}{|\mathcal{S}_{t-1}|}\right]}_{\mathcal{R}_T^{drift}}. \tag{3}$$

Let $\mathcal{R}_T^{\text{query}}$ represent the regret incurred when HYQUE actively queries for feedback. Let $\mathcal{R}_T^{\text{non-query}}$ denote the regret incurred when HYQUE decides not to query. The latter can be further decomposed into two subcomponents. (i) **Error regret ($\mathcal{R}_T^{\text{error}}$):** This arises from inaccuracies in the decision-making of HYQUE when it relies on previously gathered information about the arms without further querying. The term $\mathbb{E}\left[\max_{k \in [K]} \sum_{t \in \mathcal{S}_{t-1}} \frac{R_t^k}{|\mathcal{S}_{t-1}|}\right]$ represents the expected reward of the empirically best arm identified using feedback from $\mathcal{S}_{t-1}$. (ii) **Drift regret ($\mathcal{R}_T^{\text{drift}}$):** This accounts for the regret caused by environmental changes, such as shifts in reward distributions, that are not promptly detected due to the lack of feedback.

### 4.1.3 Bounding Total Regret

Since the on-demand query allocation converts some non-query rounds into query rounds, we first analyze the regret of HYQUE under the baseline allocation. We then prove that on-demand allocation does not increase HYQUE's regret.

**Lemma 4.9.** *With high probability, we have* $\mathcal{R}_T^{query} \leq \tilde{\mathcal{O}}\left(K^{1/3}V_T^{1/3}B^{2/3}\right)$.

For each block, as long as environmental change detection is not triggered, it suggests that significant changes are unlikely to have occurred during the query rounds (though this does not necessarily imply stability during the non-query periods). Within each block, the query rounds allocated through BAQUE, regardless of different instance scheduling strategies, exhibit similar properties to those in standard NSMAB algorithms.

**Lemma 4.10.** *With high probability, we have* $\mathcal{R}_T^{error} \leq \tilde{\mathcal{O}}\left(\frac{K^{1/3}V_T^{1/3}T}{B^{1/3}}\right)$.

If the environment remains stable and does not undergo drift, the regret caused by these inaccuracies will not exceed $\frac{2T}{B} \cdot \mathcal{R}_T^{query}$. Thus, we obtain the bound in Lemma 4.10. Finally, we analyze the regret due to environmental drift:

**Lemma 4.11.** *With high probability, we have* $\mathcal{R}_T^{drift} \leq \tilde{\mathcal{O}}\left(\frac{K^{1/3}V_T^{1/3}T}{B^{1/3}}\right)$.

By combining the bounds for $\mathcal{R}_T^{\text{error}}$ and $\mathcal{R}_T^{\text{drift}}$, we derive: $\mathcal{R}_T^{\text{non-query}} \leq \tilde{\mathcal{O}}\left(K^{1/3}V_T^{1/3}T/B^{1/3}\right)$. Substituting the bound for $\mathcal{R}_T^{\text{query}}$ and combining the bounds for $\mathcal{R}_T^{\text{query}}$ and $\mathcal{R}_T^{\text{non-query}}$, we obtain the total regret: $\mathcal{R}_T = \mathcal{R}_T^{\text{query}} + \mathcal{R}_T^{\text{non-query}} \leq \tilde{\mathcal{O}}\left(\frac{K^{1/3}V_T^{1/3}T}{B^{1/3}}\right)$. Note that the above regret bound is derived under the assumption of BAQUE. Since the on-demand allocation in HYQUE converts some non-query rounds into query rounds, the number of such converted rounds does not exceed $B/2$. Consequently, even if this conversion incurs additional regret, it remains at most of the same order as $\mathcal{R}_T^{\text{query}}$, ensuring that the overall order of the regret bound remains unaffected.

## 5 Lower Bound

**Theorem 5.1.** *Consider* CONFEE-NSMAB *with* $K \geq 2$ *arms, variation* $V_T \in \left[K^{-1}, K^{-1}B\right]$ *and constrained feedback* $B \geq K$. *For any algorithm* $\mathcal{A}$, *the following holds:*

$$\mathcal{R}_T(\mathcal{A}) \geq \Omega\left(\frac{K^{1/3}V_T^{1/3}T}{B^{1/3}}\right). \tag{4}$$

The core intuition behind the lower bound, established via a hard problem instance in which the environment changes periodically across distinct batches, is that any algorithm must consistently query arms within each batch to reliably identify the optimal arm for that period. If an algorithm uses many queries per batch to ensure high accuracy, it quickly depletes the overall query budget $B$, leaving insufficient queries for subsequent batches. Conversely, if it queries too sparsely, it fails to distinguish the optimal arm from suboptimal ones, leading to increased regret.

The lower bound highlights the fundamental challenge of allocating a limited query budget across multiple batches. We show that naive strategies—such as uniformly distributing queries or concentrating them heavily in only a few batches—lead to substantial regret. This inter-batch query allocation dilemma introduces a new layer of complexity in the NSMAB problem under constrained feedback.

*Remark* 5.2. The case of $B = T$ is particularly instructive because it aligns with the conventional NSMAB setting (where $V_T$ is typically considered within $\left[K^{-1}, K^{-1}T\right]$) and removes the feedback budget constraint. Under this $B = T$ regime, the $V_T$ range $\left[K^{-1}, K^{-1}B\right]$ specified in Theorem 5.1 conforms to the standard $\left[K^{-1}, K^{-1}T\right]$; simultaneously, the lower bound $\Omega\left(\frac{K^{1/3}V_T^{1/3}T}{B^{1/3}}\right)$ simplifies to $\Omega\left(K^{1/3}V_T^{1/3}T^{2/3}\right)$. This result aligns with established lower bound for NSMAB problem without feedback querying constraint [5]. A detailed proof is available in Appendix D.

# 6 Related Work

**Non-stationary multi-armed bandits (NSMAB).** NSMAB have been extensively studied, with a focus on adapting algorithms to environments with changing reward distributions, offering performance guarantees based on measures such as total variation or the number of abrupt changes [2, 38, 5, 15, 25, 22, 7, 3, 10]. Recent works also address smoothly evolving environments [18, 30]. Extensions to contextual bandits [33, 24, 10, 32], linear bandits [19, 41, 36], dueling bandits [27, 6, 31] and other settings [35, 40, 20, 9, 12, 23, 17, 8] have been explored. However, none of these works consider settings with a reward feedback querying constraint.

**Learning with constrained feedback.** Our work relates closely to learning with constrained feedback, where the agent manages a limited budget for acquiring observations. Prior studies [26, 13] focus on optimizing feedback allocation in stochastic settings. Other related works include MAB with paid observations [28], which assume cost and reward share units—often impractical—and require larger budgets. Bandits with additional observations [39] and knapsacks [4] either assume full reward observability or terminate when the budget is exhausted. However, these approaches focus on stationary environments, whereas it is well-known that non-stationary settings demand fundamentally different algorithmic and analytical tools.

## Limitations and Future Work

One limitation is that our HYQUE relies on a predefined query budget, which may not always align with practical applications where feedback availability can be more dynamic or influenced by external factors. Future work could explore adaptive mechanisms that adjust the query budget in real-time based on observed feedback patterns or external constraints. Additionally, while HYQUE achieves near-optimal dynamic regret guarantees in CONFEE-NSMAB setting, extending these guarantees to more complex decision-making frameworks, such as non-stationary reinforcement learning with constrained feedback, remains an open challenge. Finally, our approach primarily focuses on regret minimization, but in some applications, other performance metrics, such as fairness in feedback allocation or minimizing computational complexity, may also be important. Exploring multi-objective formulations of constrained feedback learning could provide deeper insights into balancing different performance trade-offs.

## Acknowledgements

This work was supported in part by the National Science Foundation (NSF) grants 2148309, 2315614 and 2337914, and was supported in part by funds from OUSD R&E, NIST, and industry partners as specified in the Resilient & Intelligent NextG Systems (RINGS) program. Any opinions, findings, and conclusions or recommendations expressed in this material are those of the authors and do not necessarily reflect the views of the funding agencies.

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

# A Proof of Theorem 4.1

## A.1 Total Used Queries

We prove that the total number of queries used by the algorithm cannot exceed $B$.

### A.1.1 Proof of Lemma 4.4

Let $L_0 = \lceil T/\sqrt{B} \rceil$ be the length of the consecutive non-query round sequence we are considering. Such a sequence, if attributable to the failure of initiating new instances, implies that all probabilistic attempts to start a new instance within this interval of $L_0$ rounds failed. Each new instance initiated by BAQUE would start with a query batch, thereby interrupting a sequence of non-query rounds.

The BAQUE subroutine attempts to initiate instances $\mathcal{I}_{n,m,\tau}$ at various scales $m \in \{0, 1, \ldots, n\}$ where $n$ is the time-scale parameter for the current block. If an interval of $L_0$ rounds contains any $\tau$ that is a multiple of $b \cdot 2^n$ (the block's largest scale-defining period), an instance at scale $m = n$ is initiated with probability 1, preventing the long non-query sequence. Thus, for a long non-query sequence to occur due to initiation failures, we must have $L_0 < b \cdot 2^n$. In this scenario, no initiation attempt at scale $m = n$ occurs within the $L_0$ rounds. The relevant scales for potential initiation within the interval are $m \in \{0, \ldots, \min(n - 1, \lfloor \log_2(L_0/b) \rfloor)\}$.

To establish a bound that is independent of a specific block's $n$, we consider an "effective ensemble" of scales that contribute to breaking non-query sequences. We define an effective maximum scale relevant for analyzing non-stationarity of length $L_0$. Let $M = \lfloor \log_2(T/\sqrt{B}) \rfloor$. This choice reflects a characteristic scale related to $L_0 \approx T/\sqrt{B}$. We assume that the initiation probability for an effective scale $m$ within this context can be modeled as $p'_m = 2^{(m-M)/2}$ for $m \in \{0, \ldots, M - 1\}$. (The $m = M$ scale would have $p'_M = 1$).

The probability of *not* initiating an instance at a specific opportunity $(\tau, m)$ is $1 - p'_m$. The attempts are independent. The probability that no new instance is initiated over the $L_0$ rounds is:

$$P(\text{no new instance in } L_0 \text{ rounds}) \leq \prod_{m=0}^{M-1} \prod_{\substack{\tau \in [t, t+L_0-1] \\ \tau \text{ is a multiple of } b \cdot 2^m}} (1 - 2^{(m-M)/2}).$$

Let $N_m(L_0)$ be the number of multiples of $b \cdot 2^m$ in an interval of $L_0$ rounds. $N_m(L_0) = \lfloor L_0/(b \cdot 2^m) \rfloor$. Using the inequality $1 - x \leq e^{-x}$ for $x \geq 0$:

$$P(\text{no new instance}) \leq \exp\left( -\sum_{m=0}^{M-1} N_m(L_0) \cdot 2^{(m-M)/2} \right).$$

We approximate $N_m(L_0) \approx L_0/(b \cdot 2^m)$, ignoring the floor for a lower bound on the sum in the exponent (which leads to an upper bound on the probability). For a more careful bound, $N_m(L_0) \geq L_0/(b \cdot 2^m) - 1$. Using $L_0/(b \cdot 2^m)$ directly: The sum in the exponent, $S_{\text{exp}}$, is:

$$S_{\text{exp}} \approx \sum_{m=0}^{M-1} \frac{L_0}{b \cdot 2^m} \cdot 2^{(m-M)/2} = \frac{L_0}{b \cdot 2^{M/2}} \sum_{m=0}^{M-1} 2^{-m/2}.$$

The sum $\sum_{m=0}^{M-1} 2^{-m/2} = \sum_{j=0}^{M-1} (1/\sqrt{2})^j$. As $M \to \infty$, this geometric series converges to $1/(1 - 1/\sqrt{2}) = 2 + \sqrt{2}$. For $M \geq 1$, the sum is at least 1 (for $m = 0$). Let $C_1 = \sum_{j=0}^{M-1}(1/\sqrt{2})^j$. $C_1$ is a constant factor typically between 1 and $2 + \sqrt{2}$.

Substitute $L_0 \approx T/\sqrt{B}$ and $b \approx 2T/B$:

$$\frac{L_0}{b} \approx \frac{T/\sqrt{B}}{2T/B} = \frac{B}{2\sqrt{B}} = \frac{\sqrt{B}}{2}$$

And $M = \lfloor \log_2(T/\sqrt{B}) \rfloor$, so $2^M \approx T/\sqrt{B}$ (assuming $T/\sqrt{B}$ is a power of 2 for simplicity, otherwise $2^M \leq T/\sqrt{B} < 2^{M+1}$), which implies $2^{M/2} \approx (T/\sqrt{B})^{1/2} = \sqrt{T}/B^{1/4}$. Thus,

$$S_{\text{exp}} \approx C_1 \frac{\sqrt{B}/2}{\sqrt{T}/B^{1/4}} = C_1 \frac{\sqrt{B} \cdot B^{1/4}}{2\sqrt{T}} = C_1 \frac{B^{3/4}}{2\sqrt{T}}$$

Taking $C = C_1/2$ (absorbing constants), we get $S_{\exp} \approx C\frac{B^{3/4}}{\sqrt{T}}$. Therefore, the probability that BAQUE experiences more than $L_0 = \lceil T/\sqrt{B} \rceil$ consecutive non-query rounds is at most $\exp\left(-C\frac{B^{3/4}}{\sqrt{T}}\right)$ for some universal constant $C > 0$.

*Remark* A.1. Strictly speaking, we might union-bound over all intervals $[t, t']$ of length $T/\sqrt{B}$, but the final exponent remains the same up to a constant factor when $B = \tilde{\Omega}(T^{2/3})$.

*Remark* A.2. More generally, for any $\alpha \in [0, 1]$, the probability that BAQUE experiences more than $TB^{-\alpha}$ consecutive non-query rounds within a block is at most $\exp\left(-C\frac{B^{1-\alpha/2}}{\sqrt{T}}\right)$, for some universal constant $C > 0$

### A.1.2 Proof of Lemma 4.6

We analyze the conditions under which a restart is triggered by dividing the proof into two cases, depending on the block index $n$ at the time of the restart. For ease of exposition, we assume that the time horizon $T$ is a multiple of $2B$.

**Case A: $n = 0$.** Under the algorithm's design, a restart is triggered only when a substantial shift in the reward distribution is detected. However, when $n = 0$, only a single query round is allocated within the block. This is insufficient to satisfy the detection threshold required to initiate a restart. Therefore, no restart can occur at block index $n = 0$, and the condition for a restart due to a prolonged non-query interval is not met.

**Case B: $n \geq 1$.** Now consider a restart occurring in a block with index $n' \geq 1$. Let the blocks within the corresponding phase be indexed from 1 to $n'$, each representing a distinct time scale. By the structure of the algorithm, all preceding $n' - 1$ blocks must have fully completed both their designated query and non-query rounds without triggering a restart. Due to the enforced query budget ratio, the total number of rounds in these $n' - 1$ blocks is $\frac{2T}{B}$ times the number of query rounds. On the other hand, in the $n'$-th block, the number of query rounds is at most equal to the total number of query rounds in the first $n' - 1$ blocks. Consequently, across all $n'$ blocks in the phase, the total number of rounds is at most $\frac{T}{B}$ times the number of query rounds. It follows that for any such phase in which a restart occurs, the query density must satisfy:

$$\frac{1}{2} \cdot \frac{B}{T} \leq \frac{\text{\# of query rounds in the phase}}{\text{\# of total rounds in the phase}} < \frac{B}{T}.$$

### A.1.3 Proof of Lemma 4.7

Within any phase of the algorithm, if a restart occurs at a block with index $n$, the ratio of query rounds to total rounds during that phase is bounded between $\frac{B}{2T}$ and $\frac{B}{T}$. In each phase, HYQUE maintains the query allocation within the proportion $B/T$ for every block except potentially the last one, where additional caution is needed to handle boundary effects. In the final block, to avoid a large run of consecutive query rounds following an immediate restart, HYQUE employs a "Buffer term" as discussed in Section 3.2.2. This design ensures that the query allocation does not exceed the proportion $B/T$. Specifically, the algorithm prevents excessive consecutive query rounds beyond $T/\sqrt{B}$ (Remark 4.5), $2^n$ (the query bound for the longest instance in a block), or $T - t$ (the remaining rounds). Consequently, by combining BAQUE with on-demand allocation, HYQUE ensures that the total number of queries used does not exceed $B$ over the entire time horizon $T$.

## A.2 Bounding Total Regret

### A.2.1 Regret Decomposition (Proof of Lemma 4.8)

Starting from

$$\mathcal{R}_T = \sum_{t=1}^{T}(\mu_t^* - \mu_t^k),$$

we split the time index set $[T]$ into $\mathcal{S}^{\text{query}} \cup \mathcal{S}^{\text{non-query}}$. Thus,

$$\mathcal{R}_T = \sum_{t \in \mathcal{S}^{\text{query}}}(\mu_t^* - \mu_t) + \sum_{t \in \mathcal{S}^{\text{non-query}}}(\mu_t^* - \mu_t). \tag{5}$$

The first term corresponds exactly to $\mathcal{R}_T^{\text{query}}$ since $\mu_t = \mathbb{E}[R_t]$. For the second sum, $\sum_{t \in \mathcal{S}^{\text{non-query}}}(\mu_t^* - \mu_t)$, we add and subtract $\mathbb{E}\left[\max_{k \in [K]} \sum_{t \in \mathcal{S}_{t-1}} \frac{R_t^k}{|\mathcal{S}_{t-1}|}\right]$ inside the summation:

$$\mu_t^* - \mu_t^k = \left(\mu_t^* - \mathbb{E}\left[\max_{k \in [K]} \sum_{t \in \mathcal{S}_{t-1}} \frac{R_t^k}{|\mathcal{S}_{t-1}|}\right]\right) + \left(\mathbb{E}\left[\max_{k \in [K]} \sum_{t \in \mathcal{S}_{t-1}} \frac{R_t^k}{|\mathcal{S}_{t-1}|}\right] - \mu_t\right) \quad (6)$$

We substitute (6) back into (5), which yields the final decomposition.

**Discussion.** $\mathcal{R}_T^{\text{query}}$ is straightforward: the regret incurred on rounds where actual feedback is gathered. $\mathcal{R}_T^{\text{error}}$ measures the gap between the best possible mean reward $\mu_t^*$ and the predicted reward used by the algorithm in a non-query round. $\mathcal{R}_T^{\text{drift}}$ captures how $\mathbb{E}\left[\max_{k \in [K]} \sum_{t \in \mathcal{S}_{t-1}} \frac{R_t^k}{|\mathcal{S}_{t-1}|}\right]$ deviates from the true $\mathbb{E}[R_t] = \mu_t$ because the environment changed after the last time that arm $k_t$ was observed. Note that the term $\mathbb{E}\left[\max_{k \in [K]} \sum_{t \in \mathcal{S}_{t-1}} \frac{R_t^k}{|\mathcal{S}_{t-1}|}\right]$ may instead be replaced by another proxy or predicted reward for $k_t$ based on stale or previously gathered feedback. Let $\widehat{R}_t^k$ denote this proxy reward. Then, (3) can be rewritten as:

$$\mathcal{R}_T \leq \underbrace{\sum_{t \in \mathcal{S}^{\text{query}}}\left[\mu_t^* - \mathbb{E}[R_t]\right]}_{\mathcal{R}_T^{\text{query}}} + \underbrace{\sum_{t \in \mathcal{S}^{\text{non-query}}}\left[\mu_t^* - \mathbb{E}(\widehat{R}_t^k)\right]}_{\mathcal{R}_T^{\text{error}}} + \underbrace{\sum_{t \in \mathcal{S}^{\text{non-query}}}\left[\mathbb{E}(\widehat{R}_t^k) - \mathbb{E}(R_t)\right]}_{\mathcal{R}_T^{\text{drift}}}.$$

Since on-demand allocation converts some non-query rounds into query rounds, we first analyze the regret of the algorithm under baseline allocation. Then, we prove that on-demand allocation does not increase the regret of the algorithm.

### A.2.2 Proof of Lemma 4.9

Let us begin by analyzing all the rounds where the algorithm performs a query under baseline allocation. In each block, the algorithm contains a series of instances, and for these instances, the length of the query phase is at least $2^m$. Note that for each block, as long as the environmental change detection is not triggered, it indicates that during these query rounds, the environment is unlikely to have undergone significant changes (though this does not imply that the environment remained stable during the non-query periods). Note that for the UCB1 algorithm, we have the following result.

**Lemma A.3.** *Let $\widetilde{f}_t \in [0, 1]$ denote the upper confidence bound corresponding to the optimal reward at each time step $t \in \mathcal{T}_{query}$. There exists a non-stationarity measure $V_{[1,t]}$, such that when running the* UCB1 *algorithm, for all $t \in [T]$, provided that $V_{[1,t]} \leq \sqrt{\frac{K \log t}{t}} + \frac{K}{t}$, the following holds:*

$$\widetilde{f}_t \geq \min_{\tau \in [1,t]} f_\tau^\star - V_{[1,t]},$$

$$\frac{1}{t}\sum_{\tau=1}^{t}(\widetilde{f}_\tau - R_\tau) \leq \sqrt{\frac{K \log t}{t}} + \frac{K}{t} + V_{[1,t]}.$$

*Proof.* We adapt the standard UCB1 analysis to account for a limited amount of non-stationarity $V_{[1,t]}$. Concretely, we treat the environment as "approximately stationary" up to a total variation $V_{[1,t]}$ in the optimal arm's reward. Recall that in a purely stationary $K$-armed bandit, UCB1 maintains an estimate $\widehat{\mu_\tau^k}$ of each arm's mean reward plus a confidence bonus $\text{CB}_\tau^k$ so that the *upper confidence bound* is

$$\widetilde{\mu_\tau^k} = \widehat{\mu_\tau^k} + \text{CB}_\tau^k.$$

A common choice is

$$\text{CB}_\tau^k = \sqrt{\frac{2 \log \tau}{n_\tau^k}},$$

where $n_\tau^k$ is the number of times arm $k$ has been pulled up to time $\tau$. Then the standard analysis (*cf.* [1]) shows that, for large $\tau$,

$$\widetilde{\mu_\tau^k} \geq \mu^k \quad \text{and} \quad \frac{1}{\tau}\sum_{t=1}^{\tau}\left(\widetilde{\mu_t^k} - R_t\right) \leq \sqrt{\frac{K\log\tau}{\tau}} + \mathcal{O}\left(\frac{K}{\tau}\right),$$

assuming a stationary reward distribution with mean $\mu^k$ for each arm $k$. Now suppose the reward distribution changes slowly, so that the *optimal reward* $f_t^\star = \max_k \mu_t^k$ may shift but the total variation on $[1,t]$ is bounded by $V_{[1,t]}$. In particular, $f_\tau^\star$ can differ from $f_t^\star$ by at most $\sum_{\ell=\tau}^{t-1} |f_{\ell+1}^\star - f_\ell^\star| \leq V_{[1,t]}$. We incorporate this into the UCB analysis:

- *Lower bounding $\widetilde{f}_t$.* Let $k^*$ be the best arm at some time $\tau \in [1,t]$. By stationarity analysis up to time $\tau$, $\widetilde{\mu_\tau^{k^*}}$ is a valid upper confidence bound for $\mu_\tau^{k^*}$. Then

  $$f_\tau^\star \leq \widetilde{\mu_\tau^{k^*}} \leq \widetilde{f}_t \quad \text{(since the UCB1 algorithm's bound only grows over time).}$$

  Meanwhile, $f_t^\star \geq f_\tau^\star - V_{[1,t]}$ by the definition of variation. Combining, $\widetilde{f}_t \geq f_\tau^\star \geq f_t^\star + V_{[1,t]}$, or equivalently

  $$\widetilde{f}_t \geq \min_{\tau \in [1,t]} f_\tau^\star - V_{[1,t]}.$$

  This proves the first inequality in the lemma.

- *Bounding the per-round difference $(\widetilde{f}_\tau - R_\tau)$.* In standard UCB we know that, up to $\sqrt{\frac{K\log t}{t}} + \frac{K}{t}$, the difference between the UCB and actual reward is controlled *if* the environment is effectively stationary in $[1,t]$. Since we allow a total variation $V_{[1,t]}$, the environment can shift the actual reward $R_\tau$ away from the estimated bound by at most $V_{[1,t]}$. Summation from $\tau = 1$ to $t$ yields

  $$\sum_{\tau=1}^{t}(\widetilde{f}_\tau - R_\tau) \leq \sum_{\tau=1}^{t}\left[\sqrt{\frac{K\log t}{t}} + \frac{K}{t}\right] + t\,V_{[1,t]}.$$

  Dividing by $t$ proves

  $$\frac{1}{t}\sum_{\tau=1}^{t}(\widetilde{f}_\tau - R_\tau) \leq \sqrt{\frac{K\log t}{t}} + \frac{K}{t} + V_{[1,t]}.$$

This completes the proof. $\qquad\square$

From this, we see that the UCB1 algorithm is capable of handling near-stationary environments. In contrast, Algorithm 1 follows a probabilistic scheduling mechanism that deploys multi-scale instances. Our goal is that, despite its more complex structure, it retains the same fundamental property—namely, the ability to handle near-stationary environments effectively. When addressing the non-stationary bandit problem (which, if restricted to query rounds only, reduces to a standard non-stationary bandit setting), we adopt a structure similar to the MASTER algorithm [37]. Although the introduction of baseline query allocation modifies the structure, the core analytical techniques remain comparable. By leveraging a similar proof strategy (Lemma 3 in [37]), we establish that our framework achieves the same theoretical guarantees.

Within each phase, there are different instances $\mathcal{I}_{n,m,\tau}$, where we use $\mathcal{S}_{n,m,\tau}$ to denote the set of all active rounds corresponding to each instance, parameterized by $n$ and $m$. To distinguish between query rounds and non-query rounds, let $\mathcal{S}_{n,m,\tau}^{\text{query}} = \mathcal{S}_{n,m,\tau} \cap \mathcal{S}^{\text{query}}$ represent the set of all active query rounds in instance $\mathcal{I}_{n,m,\tau}$, and let $\mathcal{S}_{n,m,\tau}^{\text{non-query}} = \mathcal{S}_{n,m,\tau} \cap \mathcal{S}^{\text{non-query}}$ denote the set of all active non-query rounds in $\mathcal{I}_{n,m,\tau}$. Then we have the following result:

**Lemma A.4.** *Let* $\hat{n} = \log_2 T + 1$, $\rho(t) = \sqrt{\frac{K\log t}{t}} + \frac{K}{t}$, *and* $\hat{\rho}(t) = 6\hat{n}\log(T/\delta)\rho(t)$. *Algorithm 1 with input* $n \leq \log_2 T$ *guarantees the following: for any instance* $\mathcal{I}_{n,m,\tau}$ *that Algorithm 1 maintains and any round index* $t \in \mathcal{S}_{n,m,\tau}^{\text{query}}$, *let* $V_{[start,t]}^{\text{query}}$ *denote the total variation over the round set*

$[start, t] \cap \mathcal{S}^{query}_{n,m,\tau}$. As long as $V^{query}_{[start,t]} \leq \rho(|\mathcal{S}^{query}_{n,m,\tau}|)$, we have, with probability at least $1 - \frac{\delta}{T}$:

$$\widetilde{g}_t \geq \min_{\tau \in [start,t]} f^\star_\tau - V^{query}_{[start,t]},$$

$$\frac{1}{t'} \sum_{\tau=start}^{t} (\widetilde{g}_\tau - R_\tau) \leq \hat{\rho}(t') + \hat{n}V^{query}_{[start,t]}.$$

Apart from the structural differences in each block, if we consider only the query rounds allocated by the baseline allocation, the HYQUE algorithm and the MASTER algorithm exhibit no fundamental differences. This aligns with our intuition that, for the CONFEE-NSMAB problem, when $B = T$, the problem effectively reduces to the non-stationary bandits setting. Consequently, for $\mathcal{R}^{query}_T$, we can derive a near-optimal regret bound for the non-stationary bandits problem as a function of $B$ rather than $T$. Therefore, the regret corresponding to query rounds can be bounded as:

$$\mathcal{R}^{query}_T \leq \widetilde{\mathcal{O}} \left( K^{1/3} V^{1/3}_T B^{2/3} \right).$$

### A.2.3  Proof of Lemma 4.10

Building upon Lemma 4.9 and 4.4, we now consider $\mathcal{R}^{error}_T$, which arises from inaccuracies in the algorithm's decision-making. In Algorithm 2, due to potential restarts, multiple phases $p = 1, 2, 3, \ldots$ may exist. Thus, we obtain:

$$\mathcal{R}^{error}_T = \sum_{t \in \mathcal{S}^{non-query}} \mathbb{E} \left[ \max_{k \in [K]} \sum_{t \in \mathcal{S}_{t-1}} \frac{R^k_t}{|\mathcal{S}_{t-1}|} \right] - \mathbb{E} \left[ \sum_{t \in \mathcal{S}^{non-query}} R_t \right]$$

$$= \sum_{p,n,m} \sum_{t \in \mathcal{S}^{non-query}_{n,m,\tau}} \mathbb{E} \left[ \max_{k \in [K]} \sum_{t \in \mathcal{S}^{query}_{n,m,\tau}} \frac{R^k_t}{|\mathcal{S}^{query}_{n,m,\tau}|} \right] - \mathbb{E} \left[ \sum_{t \in \mathcal{S}^{non-query}_{n,m,\tau}} R_t \right]$$

$$= \sum_{p,n,m} \frac{|\mathcal{S}^{non-query}_{n,m,\tau}|}{|\mathcal{S}^{query}_{n,m,\tau}|} \left( \mathbb{E} \left[ \max_{k \in [K]} \sum_{t \in \mathcal{S}^{query}_{n,m,\tau}} R^k_t \right] - \mathbb{E} \left[ \sum_{t \in \mathcal{S}^{query}_{n,m,\tau}} R_t \right] \right)$$

$$+ \left( \frac{|\mathcal{S}^{non-query}_{n,m,\tau}|}{|\mathcal{S}^{query}_{n,m,\tau}|} \mathbb{E} \left[ \sum_{t \in \mathcal{S}^{query}_{n,m,\tau}} R_t \right] - \mathbb{E} \left[ \sum_{t \in \mathcal{S}^{non-query}_{n,m,\tau}} R_t \right] \right)$$

$$\leq \frac{2|\mathcal{S}^{non-query}|}{|\mathcal{S}^{query}|} \mathcal{R}^{query}_T + \sum_{p,n,m} \left( \frac{|\mathcal{S}^{non-query}_{n,m,\tau}|}{|\mathcal{S}^{query}_{n,m,\tau}|} \mathbb{E} \left[ \sum_{t \in \mathcal{S}^{query}_{n,m,\tau}} R_t \right] - \mathbb{E} \left[ \sum_{t \in \mathcal{S}^{non-query}_{n,m,\tau}} R_t \right] \right)$$

$$\leq \frac{2T}{B} \cdot \widetilde{\mathcal{O}} \left( K^{1/3} V^{1/3}_T B^{2/3} \right) + \frac{T}{B^{2/3}} \cdot V_T$$

$$\leq \widetilde{\mathcal{O}} \left( \frac{T \cdot K^{1/3} \cdot V^{1/3}_T}{B^{1/3}} \right).$$

**Explanation of the derivation:** The *first line* defines $\mathcal{R}^{error}_T$ by summing, over every non-query round, the gap between an idealized "best average reward" (which could have been inferred from previous query feedback) and the actual reward collected. The *second line* refines this sum by grouping the non-query rounds across different phases $p$ and different multi-scale instances $(n, m, \tau)$. The *third line* factors out the ratio $\frac{|\mathcal{S}^{non-query}_{n,m,\tau}|}{|\mathcal{S}^{query}_{n,m,\tau}|}$, rewriting the maximum average reward $\max_a \sum_{t \in \mathcal{S}^{query}_{n,m,\tau}} R^k_t / |\mathcal{S}^{query}_{n,m,\tau}|$ in a form that highlights how each non-query round effectively "relies" on the estimates from the query segment. The *fourth line* then bounds the difference in these sums by relating $\max_a \sum_t R^k_t - \sum_t R_t$ to the query regret $\mathcal{R}^{query}_T$, multiplied by a factor reflecting the non-query to query ratio. In addition, it isolates a leftover term that accounts for how the environment might have shifted between the query and non-query parts. The *fifth line* substitutes known estimates for $\mathcal{R}^{query}_T$ and applies prior Lemma 4.9 and Remark A.2, yielding two main contributions on the order of $\frac{T}{B} \cdot K^{1/3} V^{1/3}_T B^{2/3}$ and $\frac{T}{B^{2/3}} \cdot V_T$. The *sixth and final line* combines these two contributions with $B = \omega(T^{3/4})$ and $V_T \leq K^{-1}\sqrt{B}$.

### A.2.4 Proof of Lemma 4.11

Finally, we analyze the regret due to environmental drift. As a result, there are at least $\sqrt{B}$ non-query segments, and each non-query segment has a length of at most $(b-1)\sqrt{B}$. Consequently, the regret due to environmental drift is bounded by:

$$
\begin{aligned}
\mathcal{R}_T^{\text{drift}} &= \sum_{t \in \mathcal{S}^{\text{non-query}}} \mu_t^* - \sum_{t \in \mathcal{S}^{\text{non-query}}} \mathbb{E}\left[\max_{k \in [K]} \sum_{t \in \mathcal{S}_{t-1}} \frac{R_t^k}{|\mathcal{S}_{t-1}|}\right] \\
&= \sum_{p,n,m} \sum_{t \in \mathcal{S}_{n,m,\tau}^{\text{non-query}}} \mu_t^* - \mathbb{E}\left[\max_{k \in [K]} \sum_{t \in \mathcal{S}_{n,m,\tau}^{\text{query}}} \frac{R_t^k}{|\mathcal{S}_{n,m,\tau}^{\text{query}}|}\right] \\
&\leq \frac{T}{B^{2/3}} \cdot V_T < \widetilde{\mathcal{O}}\left(\frac{T \cdot K^{1/3} \cdot V_T^{1/3}}{B^{1/3}}\right).
\end{aligned}
$$

**Explanation of the derivation:** In the *first line*, we define the "drift" component $\mathcal{R}_T^{\text{drift}}$ by summing, over each non-query round $t \in \mathcal{S}^{\text{non-query}}$, the gap between $\mu_t^*$ (the true best mean reward at time $t$) and the proxy-based estimate that was used. In the *second line*, we reorganize this summation by phases $p$ and multi-scale instance indices $(n, m, \tau)$, noting that non-query rounds in each instance $\mathcal{I}_{n,m,\tau}$ rely on historical query information $\mathcal{S}_{n,m,\tau}^{\text{query}}$. Finally, in the *third line*, we invoke two key facts: (i) the environment's drift between query and non-query segments can be controlled by bounding the maximal length of a non-query interval, thus contributing at most $\frac{T}{B^{2/3}} \cdot V_T$ additional regret, and (ii) under $B = \omega(T^{3/4})$ and $V_T \leq K^{-1}\sqrt{B}$, this quantity falls within $\widetilde{\mathcal{O}}\left(\frac{T K^{1/3} V_T^{1/3}}{B^{1/3}}\right)$. Hence, even if the environment may change during non-query intervals, the total extra cost is dominated by the same main order of regret.

### A.2.5 Total Regret

Combining the bounds for $\mathcal{R}_T^{\text{error}}$ and $\mathcal{R}_T^{\text{drift}}$, we obtain:

$$
\mathcal{R}_T^{\text{non-query}} \leq \widetilde{\mathcal{O}}\left(\frac{T \cdot K^{1/3} \cdot V_T^{1/3}}{B^{1/3}}\right).
$$

Substituting the bound for $\mathcal{R}_{\text{query}}$ and combining the bounds for $\mathcal{R}_T^{\text{query}}$ and $\mathcal{R}_T^{\text{non-query}}$, the total regret is given by:

$$
\mathcal{R}_T = \mathcal{R}_T^{\text{query}} + \mathcal{R}_T^{\text{non-query}} \leq \widetilde{\mathcal{O}}\left(\frac{T \cdot K^{1/3} \cdot V_T^{1/3}}{B^{1/3}}\right).
$$

This has been established under *baseline* query allocation alone. We now show that our *on-demand* mechanism—which may *convert* certain non-query rounds into query rounds (or vice versa) whenever the actual usage lags behind (or overshoots) an approximate linear pace $\frac{B}{T}$—does *not* inflate this overall regret bound. Let $\mathcal{Q}^{\text{BAQUE}}$ be the set of query rounds chosen by the baseline allocation, and let $\mathcal{Q}^{\text{ODQUE}}$ denote the additional query rounds triggered by *on-demand* scheduling. By design, on-demand scheduling only triggers new queries if the total used so far $B'$ is below $\frac{tB}{T} - \min\{\sqrt{B}, T-t\}$. Consequently, the number of such converted rounds does not exceed $B/2$, ensuring that: $\mathcal{Q}^{\text{ODQUE}} \leq B/2$. Now consider two scenarios:

- **Scenario A (Baseline only).** The algorithm uses only the baseline queries $\mathcal{Q}^{\text{BAQUE}}$, incurring regret $\widetilde{\mathcal{O}}\left(\frac{T K^{1/3} V_T^{1/3}}{B^{1/3}}\right)$.

- **Scenario B (Baseline + On-demand).** The algorithm has $\mathcal{Q}^{\text{BAQUE}} \cup \mathcal{Q}^{\text{ODQUE}}$ as query rounds. Suppose it ends up with $\mathcal{R}_T^{\text{ODQUE}}$ total regret.

Each newly allocated query round might cause partial suboptimal pulls, but a bounding analysis based on UCB1 analysis shows that $|\mathcal{Q}^{\text{ODQUE}}|$ additional query steps can only add at most $\sqrt{|\mathcal{Q}^{\text{ODQUE}}| \cdot K \log T}$ to the query-phase regret. Because $|\mathcal{Q}^{\text{ODQUE}}| \leq B/2$, this is at most on

the order of the baseline query regret $\mathcal{R}_T^{\text{query}}$ or smaller. Meanwhile, those same converted rounds *lower* the non-query portion; effectively, the error from stale estimates in scenario B should be $\leq$ that in scenario A.

Consequently, even if this conversion incurs additional regret, the difference is at most a constant factor times $\mathcal{R}_T^{\text{query}}$, which does not affect the overall order of the regret bound.

# B  Algorithm for Known Variation Setting

## B.1  The Rexp3B Algorithm

---
**Algorithm 3** Rexp3B: Rexp3 algorithm with Budgeted feedback

---
1: **Inputs**: Feedback budget $B$, time horizon $T$, variation budget $V_T$, and number of arms $K$.
2: Set $\Delta_T = \frac{T \cdot (K \log K)^{1/3}}{B^{1/3} \cdot V_T^{2/3}}$, $\Delta_B = \lfloor \frac{B}{T} \cdot \Delta_T \rfloor$, and $\gamma = \min \left\{ 1, \sqrt{\frac{K \log K}{(e-1)\Delta_B}} \right\}$.
3: Initialize $w_t^k = 1$ for all $k \in [K]$.
4: **for** $j = 1, \ldots, \lceil T/\Delta_T \rceil$ **do**
5:     Set the current batch's start time $\tau = (j-1)\Delta_T$.
6:     **for** $t = \tau + 1, \ldots, \min\{T, \tau + \Delta_B\}$ **do**
7:         For each $k \in [K]$, set

$$p_t^k = (1 - \gamma) \frac{w_t^k}{\sum_{k'=1}^{K} w_t^{k'}} + \frac{\gamma}{K}$$

8:         Draw an arm $k' \in [K]$ according to the distribution $\{p_t^k\}_{k=1}^{K}$
9:         Receive a reward $R_t^{k'}$
10:        For $k'$, set $\hat{X}_t^{k'} = R_t^{k'}/p_t^{k'}$, and for any $a \neq k'$ set $\hat{X}_t^k = 0$
11:        For all $k \in [K]$, update:

$$w_{t+1}^k = w_t^k \exp\left( \frac{\gamma \hat{X}_t^k}{K} \right)$$

12:     **for** $t = \tau + \Delta_B + 1, \ldots, \min\{T, \tau + \Delta_T\}$ **do**
13:         Select an arm uniformly at random from the set $\{a_{t'}' : t' = \tau + 1, \ldots, \tau + \Delta_B\}$.

---

The Rexp3B algorithm operates by dividing the time horizon $T$ into discrete batches, dynamically adjusting its query strategy based on the allocated feedback budget and the known variation budget $V_T$. Initially, the time horizon $T$ is partitioned into batches of size $\Delta_T$, ensuring a balanced allocation of resources across time. Within each batch, the algorithm allocates a subset of $\Delta_B$ steps to a query phase. During this phase, it employs the EXP3 strategy to select arms and actively request feedback, prioritizing arms with higher estimated rewards by assigning them greater selection probabilities. After the query phase, the algorithm transitions to a non-query phase for the remaining $\Delta_T - \Delta_B$ steps of the batch. In this phase, the algorithm selects arms without requesting additional feedback, relying instead on the information gathered during the query phase. Specifically, it uniformly randomly selects an arm from the set of arms chosen during the current batch's query phase. In practice, this sampling strategy could be replaced by others, such as greedily selecting the arm with the highest weight from the query phase.

To achieve the desired regret bounds, the algorithm parameters are carefully configured, including the batch size $\Delta_T$, the learning rate $\gamma$, and the number of query steps per batch $\Delta_B$. These parameter settings enable Rexp3B to balance query and non-query steps while strictly adhering to the feedback budget. Furthermore, the algorithm accommodates the non-stationary nature of the environment by adjusting its behavior in accordance with the known variation budget $V_T$. As a result, the Rexp3B algorithm is theoretically guaranteed to achieve the desired regret bounds, providing an effective solution for multi-armed bandit problems characterized by budgeted feedback and non-stationary rewards.

**Theorem B.1** (Regret Bound for Rexp3B). *Under the known variation setting with variation $V_T \in [K^{-1}, K^{-1}B]$, the Rexp3B algorithm uses at most $B$ query rounds and satisfies the following regret*

*bound:*

$$\mathcal{R}_T \leq \tilde{\mathcal{O}}\left(\frac{K^{1/3}V_T^{1/3}T}{B^{1/3}}\right).$$

## B.2 Regret Analysis (Proof of Theorem B.1)

As discussed in Section 4.1.2, the regret $\mathcal{R}_T$ can be expressed as:

$$\mathcal{R}_T = \mathcal{R}_T^{\text{query}} + \mathcal{R}_T^{\text{non-query}} = \mathcal{R}_T^{\text{query}} + \mathcal{R}_T^{\text{error}} + \mathcal{R}_T^{\text{drift}}.$$

The term $\mathcal{R}_T^{\text{query}}$ represents the regret incurred during the time steps when the algorithm actively queries for feedback. The term $\mathcal{R}_T^{\text{non-query}}$ captures the regret incurred when the algorithm decides not to query for feedback. This component can be further decomposed into two subcomponents. The first, $\mathcal{R}_T^{\text{error}}$, arises from inaccuracies in the algorithm's decision-making when it relies on previously gathered information about the arms without further querying. The second subcomponent, $\mathcal{R}_T^{\text{drift}}$, accounts for the regret caused by environmental changes, such as shifts in the reward distributions, that are not promptly detected due to the absence of feedback.

### Step1: Bounding $\mathcal{R}_T^{\text{query}}$

The term $\mathcal{R}_T^{\text{query}}$ is the regret arising from the $\Delta_B$ query rounds in each batch. Fix $T \geq 1$, $K \geq 2$, and $V_T \in [K^{-1}, K^{-1}B]$. We partition the time horizon $T$ into a sequence of batches $\mathcal{T}_1, \ldots, \mathcal{T}_m$ of size $\Delta_T$ each (except possibly the last batch). Let $\mathcal{T}_j^{\text{query}}$ denote the set of query rounds in batch $\mathcal{T}_j, j \in \{1, \ldots, m\}$. We decompose the regret $\mathcal{R}_T^{\text{query}}$ as:

$$\mathcal{R}_T^{\text{query}} = \sum_{t \in \mathcal{S}^{\text{query}}} \mu_t^* - \mathbb{E}\left[\sum_{t \in \mathcal{S}^{\text{query}}} R_t\right] = \sum_j \mathbb{E}\left[\sum_{t \in \mathcal{T}_j^{\text{query}}} (\mu_t^* - \mu_t)\right]$$

$$= \sum_j \underbrace{\sum_{t \in \mathcal{T}_j^{\text{query}}} \mu_t^* - \mathbb{E}\left[\max_{k \in [K]} \sum_{t \in \mathcal{T}_j^{\text{query}}} R_t^k\right]}_{J_{1,j}} + \underbrace{\mathbb{E}\left[\max_{k \in [K]} \sum_{t \in \mathcal{T}_j^{\text{query}}} R_t^k\right] - \mathbb{E}\left[\sum_{t \in \mathcal{T}_j^{\text{query}}} R_t\right]}_{J_{2,j}}. \tag{7}$$

Here, $J_{1,j}$ is the expected loss associated with using a single action over batch $j$, and $J_{2,j}$ is the expected regret relative to the best static action in batch $j$. Let $V_j$ denote the total variation in expected rewards during batch $j$:

$$V_j = \sum_{t \in \mathcal{T}_j} \max_{k \in [K]} |\mu_{t+1}^k - \mu_t^k|.$$

We note that:

$$\sum_{j=1}^m V_j = \sum_{j=1}^m \sum_{t \in \mathcal{T}_j} \max_{k \in [K]} |\mu_{t+1}^k - \mu_t^k| \leq V_T.$$

Let $k_j$ be an arm with the best expected performance over $\mathcal{T}_j^{\text{query}}$:

$$k_j \in \arg\max_{k \in [K]} \left\{\sum_{t \in \mathcal{T}_j^{\text{query}}} \mu_t^k\right\}.$$

Then,

$$\max_{k \in [K]} \left\{\sum_{t \in \mathcal{T}_j^{\text{query}}} \mu_t^k\right\} = \sum_{t \in \mathcal{T}_j^{\text{query}}} \mu_t^{k_j} = \mathbb{E}\left[\sum_{t \in \mathcal{T}_j^{\text{query}}} R_t^{k_j}\right] \leq \mathbb{E}\left[\max_{k \in [K]} \left\{\sum_{t \in \mathcal{T}_j^{\text{query}}} R_t^k\right\}\right],$$

For term $J_{1,j}$, we have:

$$J_{1,j} = \sum_{t \in \mathcal{T}_j^{\text{query}}} \mu_t^* - \mathbb{E}\left[\max_{k \in [K]} \left\{\sum_{t \in \mathcal{T}_j^{\text{query}}} R_t^k\right\}\right] \leq \sum_{t \in \mathcal{T}_j^{\text{query}}} \left(\mu_t^* - \mu_t^{k_j}\right) \leq 2V_j\Delta_B. \tag{8}$$

For term $J_{2,j}$, using the standard regret bound for the EXP3 algorithm in adversarial settings, we have:

$$J_{2,j} = \mathbb{E}\left[\max_{k\in[K]}\sum_{t\in\mathcal{T}_j^{\text{query}}} R_t^k\right] - \mathbb{E}\left[\sum_{t\in\mathcal{T}_j^{\text{query}}} R_t\right] \leq 2\sqrt{(e-1)\Delta_B K \log K}. \tag{9}$$

We substitute (8) and (9) into (7) and sum over all $m = \left\lceil \frac{T}{\Delta_T} \right\rceil$ batches. This yields the total regret incurred during all query rounds:

$$\mathcal{R}_T^{\text{query}} \leq \sum_{j=1}^m \left(2\sqrt{e-1}\sqrt{\Delta_B K \log K} + 2V_j \Delta_B\right)$$

$$\leq \left(\frac{T}{\Delta_T} + 1\right) \cdot 2\sqrt{e-1}\sqrt{\Delta_B K \log K} + 2\Delta_B V_T.$$

**Step2: Bounding $\mathcal{R}_T^{\text{non-query}}$**

In the remaining $\Delta_T - \Delta_B$ steps of each batch (denote them by $\mathcal{T}_j^{\text{non-query}}$), the algorithm *does not* request feedback. The main concern here is that the environment might "drift" substantially in these non-query steps without being detected. We start with the term $\mathcal{R}_T^{\text{error}}$. We have:

$$\mathcal{R}_T^{\text{error}} = \sum_{t\in\mathcal{S}^{\text{non-query}}} \mathbb{E}\left[\max_{k\in[K]}\sum_{t\in\mathcal{S}_{t-1}}\frac{R_t^k}{|\mathcal{S}_{t-1}|}\right] - \mathbb{E}\left[\sum_{t\in\mathcal{S}^{\text{non-query}}} R_t\right]$$

$$= \sum_{j=1}^m \sum_{t\in\mathcal{T}_j^{\text{non-query}}} \mathbb{E}\left[\max_{k\in[K]}\sum_{t\in\mathcal{T}_j^{\text{query}}}\frac{R_t^k}{|\mathcal{T}_j^{\text{query}}|}\right] - \mathbb{E}\left[\sum_{t\in\mathcal{T}_j^{\text{non-query}}} R_t\right]$$

$$= \sum_{j=1}^m \frac{|\mathcal{T}_j^{\text{non-query}}|}{|\mathcal{T}_j^{\text{query}}|}\underbrace{\left(\mathbb{E}\left[\max_{k\in[K]}\sum_{t\in\mathcal{T}_j^{\text{query}}} R_t^k\right] - \mathbb{E}\left[\sum_{t\in\mathcal{T}_j^{\text{query}}} R_t\right]\right)}_{J_{2,j}}$$

$$+ \left(\frac{|\mathcal{T}_j^{\text{non-query}}|}{|\mathcal{T}_j^{\text{query}}|}\mathbb{E}\left[\sum_{t\in\mathcal{T}_j^{\text{query}}} R_t\right] - \mathbb{E}\left[\sum_{t\in\mathcal{T}_j^{\text{non-query}}} R_t\right]\right)$$

$$< 2\sum_{j=1}^m \frac{\Delta_T - \Delta_B}{\Delta_B}\sqrt{(e-1)\Delta_B K \log K} + V_j\Delta_T$$

$$\leq 2T\sqrt{\frac{(e-1)K\log K}{\Delta_B}} + 2V_T\Delta_T$$

**Explanation of the derivation:** We decompose the error term $\mathcal{R}_T^{\text{error}}$ by initially defining it (Line 1) as the sum over non-query rounds of the difference between a hypothetical "best average reward" inferred from prior query feedback and the actual total reward in those non-query rounds. We then reorganize this sum by batches $j = 1, \ldots, m$ (Line 2), where each batch has a query part $\mathcal{T}_j^{\text{query}}$ and a non-query part $\mathcal{T}_j^{\text{non-query}}$ (Line 3). At this stage, the bracketed term $J_{2,j}$ compares the "best possible reward sums" in query rounds to the algorithm's chosen sums, bounded by the adversarial regret argument. Substituting known estimates yields the expression in Line 4, where the ratio $\frac{\Delta_T - \Delta_B}{\Delta_B}$ reflects how many non-query rounds depend on each query segment, and $V_j$ captures environment

variation within that batch. Then we consider $\mathcal{R}_T^{\text{drift}}$:

$$
\begin{aligned}
\mathcal{R}_T^{\text{drift}} &= \sum_{t \in \mathcal{S}^{\text{non-query}}} \mu_t^* - \sum_{t \in \mathcal{S}^{\text{non-query}}} \mathbb{E}\left[\max_{k \in [K]} \sum_{t \in \mathcal{S}_{t-1}} \frac{R_t^k}{|\mathcal{S}_{t-1}|}\right] \\
&= \sum_{j=1}^{m} \sum_{t \in \mathcal{T}_j^{\text{non-query}}} \mu_t^* - \mathbb{E}\left[\max_{k \in [K]} \sum_{t \in \mathcal{T}_j^{\text{query}}} \frac{R_t^k}{|\mathcal{T}_j^{\text{query}}|}\right] \\
&\leq 2 \sum_{j=1}^{m} (\Delta_T - \Delta_B) V_j \\
&\leq 2 \left(\frac{T}{\Delta_T} + 1\right) (\Delta_T - \Delta_B) \frac{V_T \cdot \Delta_T}{T} \\
&\leq 2 (T + \Delta_T) \frac{V_T \Delta_T}{T} \leq 4 V_T \Delta_T.
\end{aligned}
$$

**Explanation of the derivation:** We similarly define the drift term $\mathcal{R}_T^{\text{drift}}$ (Line 1) as the gap between the true best mean reward $\mu_t^*$ in each non-query round and the "best average feedback" constructed from query data. We again partition by batches (Line 2) and argue (Line 3) that each batch's drift can be bounded by the product of its non-query length $(\Delta_T - \Delta_B)$ and a local measure of variation $V_j$. Accumulating over batches and simplifying (Line 4–6) shows that the drift contribution is also capped, typically at $\mathcal{O}(V_T \Delta_T)$.

Therefore, the regret during the exploitation phase can be bounded as:

$$
\mathcal{R}_T^{\text{non-query}} = \mathcal{R}_T^{\text{error}} + \mathcal{R}_T^{\text{drift}} \leq 6 V_T \Delta_T + 2T \sqrt{\frac{(e-1)K \log K}{\Delta_B}}.
$$

**Step3: Combining and Minimizing Over $\Delta_T$**

The total regret could be bounded as:

$$
\begin{aligned}
\mathcal{R}_T &= \mathcal{R}_T^{\text{query}} + \mathcal{R}_T^{\text{non-query}} \\
&= \left(\frac{T}{\Delta_T} + 1\right) \cdot 2\sqrt{e-1} \sqrt{\Delta_B K \log K} + 2\Delta_B V_T + 6 V_T \Delta_T + 2T \sqrt{\frac{(e-1)K \log K}{\Delta_B}}.
\end{aligned}
$$

By choosing the batch size $\Delta_T$ as

$$
\Delta_T = \frac{T \cdot (K \log K)^{1/3}}{B^{1/3} \cdot V_T^{2/3}},
$$

we minimize the cumulative regret to:

$$
\mathcal{R}_T \leq \mathcal{O}\left(\frac{T \cdot (K \log K)^{1/3} \cdot V_T^{1/3}}{B^{1/3}}\right).
$$

## C  Extensions

In this section, we show that our framework naturally accommodates more general non-stationary learning problems, including contextual bandits, linear bandits, and certain reinforcement learning (RL) scenarios. Concretely, one only needs to *replace* the base algorithm (UCB1 in our default BAQUE) with a corresponding algorithm ALG *suited to* the target setting (*e.g.*, a linear bandit algorithm or a contextual bandit procedure).

### C.1  Extended Baseline Allocation: BAQUE with ALG

**Algorithm 4 explanation.**  Compared with Algorithm 1 in Section 1, we replace UCB1 by a generic base algorithm ALG. The $\rho(\cdot)$ and $\frac{\rho(2^n)}{\rho(2^m)}$ criteria follow from Assumption C.2, ensuring the *instance*

---

**Algorithm 4** Extended BAQUE for General Base Algorithm ALG

---

**Require:** Query budget ratio $b = \lceil 2T/B \rceil$, time-scale parameter $n$, non-increasing function $\rho(\cdot)$.

1: **for** $\tau = 0, b-1, 2b-1, \ldots, 2^n \cdot b - 1$ **do**
2:     **for** $m = n, n-1, \ldots, 0$ **do**
3:         **if** $\tau$ is a multiple of $b \cdot 2^m$ **then**
4:             With probability $\frac{\rho(2^n)}{\rho(2^m)}$, initiate a new instance $\mathcal{I}_{n,m,\tau}$ spanning rounds $[\tau+1, \tau + b \cdot 2^m]$;
5: **for** each instance $\mathcal{I}_{n,m,\tau}$ **do**
6:     Let $\mathcal{S}_{n,m,\tau}$ be its *active* rounds;
7:     **Query batch**: For the first $\max(1, \lfloor |\mathcal{S}_{n,m,\tau}|/b \rfloor)$ active rounds, run ALG, collecting rewards and updating the index $\tilde{g}_t$, which is $\tilde{f}_t$ in Assumption C.2.
8:     **Non-query batch**: In the remaining active rounds, pick arms according to their frequencies from query batch (no reward feedback).

---

*scheduling probabilities* remain consistent with the required non-stationarity measure $V$ and error margin.

*Remark* C.1. Figure 1 (earlier) remains unchanged: we still have multi-scale instances, each with a query batch and a non-query batch. Only the internal UCB1 logic is replaced by ALG, which might be, *e.g.*, a linear bandit algorithm or a contextual bandit procedure.

## C.2   Extended Hybrid Framework: HYQUE with ALG

---

**Algorithm 5** Extended HYQUE

---

1: **Initialize:** current round $t \leftarrow 1$, used queries $B' \leftarrow 0$.
2: **for** $n = 0, 1, \ldots$ **do**
3:     Set $t_n \leftarrow t$ and initialize BAQUE for block $[t_n, t_n + b \cdot 2^n - 1]$ with the time-scale parameter $n$;
4:     **while** $t < t_n + b \cdot 2^n$ **do**
5:         **if** current instance has queries **then**
6:             Receive index $\tilde{g}_t$ and selected arm from BAQUE, play it, increment $t$ and $B'$;
7:             Update BAQUE instance with the observed feedback;
8:             Perform environmental change tests. If any test fails, restart a new phase from *Line 2*;
9:         **else**
10:             **if** $B' < \frac{t \cdot B}{T} - \min\{T/\sqrt{B}, 2^n, T - t\}$ **then**
11:                 Convert the current non-query round into a query round and jump to *Line 6*;
12:             No feedback requested; $t \leftarrow t + 1$.

---

**Algorithm 5 explanation.** This is a direct generalization of Algorithm 2 in the main text, except we embed ALG inside BAQUE. The environment-change tests remain the same, or adapt to ALG's specific estimates of reward. The on-demand logic remains: if the total used queries $B'$ is far below $\frac{t B}{T}$, we convert that round into a query.

## C.3   General Conditions on the Base Algorithm ALG

The following assumption from Wei and Luo [37] ensure that ALG yields upper confidence estimates for the optimal reward under bounded non-stationarity.

**Assumption C.2** (37). ALG outputs an auxiliary quantity $\tilde{f}_t \in [0, 1]$ at each time step $t$. There exists a non-stationarity measure $V$ and a non-increasing function $\rho : [T] \to \mathbb{R}$ such that when running ALG, the following conditions hold for all $t \in [T]$ provided that $V_{[1,t]} \le \rho(t)$:

$$\tilde{f}_t \ge \min_{\tau \in [1,t]} f_\tau^\star - V_{[1,t]}, \qquad \frac{1}{t} \sum_{\tau=1}^{t} (\tilde{f}_\tau - r_\tau) \le \rho(t) + V_{[1,t]}.$$

This guarantees hold with probability at least $1 - \frac{\delta}{T}$. Additionally, $\rho(t) \geq \frac{1}{\sqrt{t}}$ and $C(t) = t\rho(t)$ is a non-decreasing function.

This assumption ensures that the base algorithm (ALG) provides reliable estimates of the optimal reward and maintains a controlled discrepancy between the auxiliary outputs and the actual rewards. The function $\rho(t)$ captures the uncertainty or error margin, which decreases over time.

**Implication.** Under Assumption C.2, the entire analysis remains the same, simply replacing UCB1 with ALG, and ensuring the scheduling probabilities scale via $\rho(\cdot)$ as indicated. This yields a near-optimal dynamic regret for a broader class of non-stationary settings, matching the idea of the MASTER algorithm [37], but now *budgeted* by $B$ queries thanks to our hybrid query method.

## D  Proof of Theorem 5.1

We construct a family of problem instances and analyze the regret that any algorithm must incur on these instances.

### Step 1: Batching the Horizon and Constructing Reward Means

Partition the time horizon $T$ into $m = \lceil \frac{T}{\Delta} \rceil$ batches, each of size $\Delta$ (except possibly the last). Denote

$$\mathcal{T}_j = \{t : (j-1)\Delta + 1 \leq t \leq \min\{j\Delta, T\}\}, \quad j = 1, \ldots, m.$$

We will choose $\Delta$ later to balance the terms in the lower bound. For a small $0 < \varepsilon \leq \frac{1}{4}$ (also determined later), define a family $\mathcal{V}'$ of reward sequences such that:

- $\mu_t^k \in \left\{ \frac{1}{2}, \frac{1}{2} + \varepsilon \right\}$ for all $k \in [K]$, $t \in [T]$;
- In each batch $\mathcal{T}_j$, exactly one "good" arm $k_j$ has mean $\frac{1}{2} + \varepsilon$, while all others remain at $\frac{1}{2}$;
- These means are constant within each batch (no within-batch variation).

The total variation over time for any $\mu \in \mathcal{V}'$ is:

$$\sum_{t=1}^{T-1} \sup_k \left| \mu_t^k - \mu_{t+1}^k \right| = (m-1)\varepsilon \leq \frac{T\varepsilon}{\Delta}.$$

Hence if we set

$$\varepsilon \leq \frac{V_T \Delta}{T},$$

then $\mathcal{V}' \subseteq \mathcal{V}$, ensuring all such sequences are valid under the variation budget $V_T$.

### Step 2: Single-Batch Analysis under a Feedback Budget

In each batch $\mathcal{T}_j$, we analyze the expected regret under the constraint of the corresponding query budget $B_j$ allocated to this batch. For any algorithm $\mathcal{A}$, define:

- $n_j(k) = \mathbb{E}\big[(\text{\# times arm } k \text{ is pulled in batch } j)\big]$;
- $n_j^q(k) = \mathbb{E}\big[(\text{\# times feedback of arm } k \text{ is actually observed in batch } j)\big]$.

Because the total feedback across *all* batches is at most $B$, we have

$$\sum_{j=1}^m \sum_{k \in [K]} \mathbb{E}[n_j^q(k)] \leq \sum_{j=1}^m B_j \leq B.$$

We aim to lower bound the regret in each batch $\mathcal{T}_j$, accounting for the limited feedback. Let $\nu$ be an instance with all arms having mean $\frac{1}{2}$, and let $\nu'$ differ only by giving arm $k_j$ a mean of $(\frac{1}{2} + \varepsilon)$ in batch $j$. Let $\mathbb{P}_\nu$ and $\mathbb{P}_{\nu'}$ be the probability measures (over entire batch's observation process) under $\nu$ and $\nu'$, respectively. The Kullback-Leibler divergence satisfies

$$\mathrm{KL}(\mathbb{P}_\nu, \mathbb{P}_{\nu'}) \leq \mathbb{E}_\nu\big[n_j^q(k_j)\big] \cdot \mathrm{KL}\left(\tfrac{1}{2}, \tfrac{1}{2} + \varepsilon\right).$$

Here $\mathrm{KL}\left(\frac{1}{2}, \frac{1}{2} + \varepsilon\right) \leq 2\varepsilon^2$ for $\varepsilon \leq \frac{1}{4}$. By Pinsker's inequality,

$$\left|\mathbb{E}_\nu[f] - \mathbb{E}_{\nu'}[f]\right| \leq \|f\|_\infty \sqrt{\tfrac{1}{2}\,\mathrm{KL}(\mathbb{P}_\nu, \mathbb{P}_{\nu'})},$$

where $\|f\|_\infty$ is the maximum absolute value of $f$. Taking $f = n_j(k_j)$, which is clearly at most $\Delta$ in magnitude (the arm cannot be pulled more than $\Delta$ times in that batch), we get

$$\left|\mathbb{E}_\nu[n_j(k_j)] - \mathbb{E}_{\nu'}[n_j(k_j)]\right| \leq \Delta\sqrt{\tfrac{1}{2}\,\mathrm{KL}(\mathbb{P}_\nu, \mathbb{P}_{\nu'})} \leq \Delta\varepsilon\sqrt{2\,\mathbb{E}_\nu[n_j^q(k_j)]}.$$

Then we have

$$\mathbb{E}_{\nu'}[n_j(k_j)] \leq \mathbb{E}_\nu[n_j(k_j)] + \Delta\,\varepsilon\sqrt{2\,\mathbb{E}_\nu[n_j^q(k_j)]}. \tag{10}$$

To further bound $\mathbb{E}_{\nu'}[n_j(k_j)]$, we use the following result:

**Lemma D.1** (Efroni et al. [13])**.** *Let $x, y \in \mathbb{R}_+^n$ for some $n \geq 2$, and assume that $\sum_{i=1}^n x_i \leq X$ and $\sum_{i=1}^n y_i \leq Y$. Then, for any $\alpha \in (1, 2)$ and $\beta \geq \frac{3\alpha}{\alpha-1}$, there exists an index $k \in [n]$ such that $x_k \leq \frac{\alpha X}{n}$ and $y_k \leq \frac{\beta Y}{n}$ simultaneously.*

Let $\alpha = \frac{5}{4}$, $\beta = 15$, $x_k = \mathbb{E}_\nu[n_j(k)]$ and $y_k = \mathbb{E}_\nu[n_j^q(k)]$ for $k \in [K]$. Then,

$$\sum_{k \in [K]} x_k \leq \Delta, \quad \sum_{k \in [K]} y_k \leq B_j.$$

By Lemma D.1, there exists an arm $k_j$ such that:

$$x_{k_j} \leq \frac{\alpha\Delta}{K} = \frac{5\Delta}{4K}, \quad y_{k_j} \leq \frac{\beta B_j}{K} = \frac{15 B_j}{K}.$$

Substituting the above two terms into (10), we have

$$\mathbb{E}_{\nu'}[n_j(k_j)] \leq \frac{5\Delta}{4K} + \Delta\varepsilon\sqrt{\frac{15 B_j}{K}}.$$

Let $\mathcal{R}_j$ denote the expected regret in batch $j$. Under $\nu'$, the expected regret in batch $j$ is at least:

$$\mathcal{R}_j \geq \varepsilon\left(\Delta - \mathbb{E}_{\nu'}[n_j(k_j)]\right).$$

Therefore,

$$\mathcal{R}_j \geq \varepsilon\left(\Delta - \frac{5\Delta}{4K} - \Delta\varepsilon\sqrt{\frac{30 B_j}{K}}\right) = \Delta\varepsilon\left(1 - \frac{5}{4K} - \varepsilon\sqrt{\frac{30 B_j}{K}}\right). \tag{11}$$

**Step 3: Summing Regret over $m$ Batches**

We have $m = \lceil T/\Delta\rceil$ batches in total. Summing (11) over $j = 1, \ldots, m$ yields:

$$\begin{aligned}
\mathcal{R}_T(\mathcal{A}) = \sum_{j=1}^m \mathcal{R}_j &\geq \sum_{j=1}^m \Delta\varepsilon\left(1 - \frac{5}{4K} - \varepsilon\sqrt{\frac{30 B_j}{K}}\right) \\
&\geq (m-1)\Delta\varepsilon\left(1 - \frac{5}{4K} - \varepsilon\sqrt{\frac{30 B\Delta}{KT}}\right) \\
&\geq \frac{1}{2}T\varepsilon\left(1 - \frac{5}{4K} - \varepsilon\sqrt{\frac{30 B\Delta}{KT}}\right).
\end{aligned} \tag{12}$$

We choose $\varepsilon$ and $\Delta$ such that:

$$\varepsilon = \frac{V_T\Delta}{T}, \quad \varepsilon\sqrt{\frac{30 B\Delta}{KT}} = \frac{1}{8}.$$

Recall that $\varepsilon = \frac{V_T\Delta}{T}$. Substituting, we have:

$$\frac{V_T\Delta}{T}\sqrt{\frac{30 B\Delta}{KT}} = \frac{1}{8}.$$

Therefore,

$$\Delta = \left( \frac{KT^3}{1920V_T^2B} \right)^{1/3}, \quad \varepsilon = \frac{V_T\Delta}{T} = \frac{V_T}{T} \left( \frac{KT^3}{1920V_T^2B} \right)^{1/3} = \frac{K^{1/3}V_T^{1/3}}{1920^{1/3}B^{1/3}}.$$

Since $K \geq 2$, $\frac{5}{4K} \leq \frac{5}{8}$, so:

$$1 - \frac{5}{4K} - \frac{1}{8} \geq 1 - \frac{5}{8} - \frac{1}{8} = \frac{1}{4}.$$

We substitute these $\Delta, \varepsilon$ back into (12):

$$\mathcal{R}_T(\mathcal{A}) \geq \frac{T\varepsilon}{8} = \frac{T}{8} \left( \frac{K^{1/3}V_T^{1/3}}{1920^{1/3}B^{1/3}} \right) = \frac{1}{8 \cdot 1920^{1/3}} \frac{K^{1/3}V_T^{1/3}T}{B^{1/3}}.$$

This shows that any algorithm $\mathcal{A}$ limited to $B$ queries suffers $\Omega\left( \frac{K^{1/3}V_T^{1/3}T}{B^{1/3}} \right)$ regret in the worst case, completing the proof.

# E  Experiment Results

In this section, we compare different query allocation strategies. We base our comparison on the `Rexp3B-Sample` algorithm (Appendix B), which assumes a known variation budget $V_T$, as its structure is more easily adapted to accommodate different allocation policies.

**Environments.**  We evaluate all algorithms across three distinct non-stationary environments. In all settings, the time horizon is set to $T = 200,000$, the number of arms is $K = 5$, and all non-optimal arms provide a baseline reward of $\mu_{\text{base}} = 0.5$. First, we use a standard Piecewise Stationary environment, where the time horizon is divided into 40 equal epochs. In each epoch, a single optimal arm's reward is elevated, and this optimal arm cycles deterministically at each change point. This setting, with a query budget of $B = 100,000$ and total variation $V_T = 20$. We also test on a Random Changepoints environment with randomly distributed change points, and a Low Query Budget environment where the budget is reduced to $B = 20,000$. All runs are executed on a machine equipped with a 12th Gen Intel(R) Core(TM) i9-12900HX processor.

**Baselines.**  We compare our main algorithm, `Rexp3B-Sample` with frequency-based sampling in non-query phase, against four baselines. The `Rexp3B-Greedy` baseline follows the same batched structure but commits to a purely greedy strategy in non-query phase. The `Uniform Query` baseline also uses a batched approach but distributes its queries uniformly within each batch. The `Explore-Then-Commit (ETC)` baseline is a classic strategy that expends its entire query budget in an initial exploration phase before committing to a fixed policy. Finally, the `Random` baseline selects arms uniformly at random, serving as a performance lower bound.

**Results.**  Figure 3 presents the cumulative regret of all algorithms across the three environments. Across all settings, the `Rexp3B-Greedy` strategy consistently achieves the best performance. In contrast, `Uniform Query` performs poorly, due to its delayed exploration within each batch. The `Rexp3B-Sample` algorithm delivers a middle-ground performance, appearing conservative in these non-adversarial environments. The `Explore-Then-Commit (ETC)` strategy performs well during its initial query phase but degrades to random-like performance after its budget is exhausted.

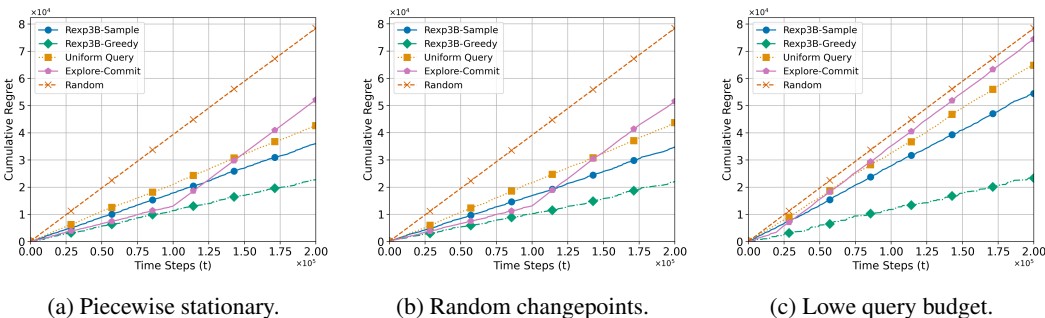

(a) Piecewise stationary.

(b) Random changepoints.

(c) Lowe query budget.

Figure 3: Regret with different non-statioanry environments.

