# OpenReview forum: "Constrained Feedback Learning for Non-Stationary Multi-Armed Bandits"
_NeurIPS.cc/2025/Conference — NeurIPS 2025 poster_

### Official Review · Reviewer_ndtv · 2025-06-23

**Clarity:** 3
**Significance:** 2
**Originality:** 2
**Rating:** 4
**Confidence:** 3

**Summary:**

This paper studies non-stationary multi-armed bandits (NSMAB) under the constrained feedback setting, where the reward feedback is limited. Given a query budget $B$,  the total number of reward feedback queries available to the agent over the time horizon $T$ is limited by a reward query budget $B$. This problem is named the constrained feedback in non-stationary multi-armed bandits (CONFEE-NSMAB)
There are two main focuses: "How does the query budget constraint affect the problem difficulty?" and "Can we construct an algorithm that works well in CONFEE-NSMAB without knowing the degree of non-stationarity?".
The authors introduced a nearly-optimal algorithm that does not require any knowledge of the degree of non-stationarity.

**Questions:**

- If we do not know anything about the non-stationarity, what is the point of "detecting" environmental changes?
- Is deriving a distribution-dependent regret bound too hard for a non-stationary bandit?
- Even when we know $V_{T}$, this algorithm can be applied to the non-stationary bandit. Isn't it necessary to compare the proposed algorithm with existing algorithms for cases when we know $V_{T}$? Conversely, I think it is valuable to compare the proposed algorithm with existing algorithms with $V_{T}=T$.

**Ethical Concerns:**

["NO or VERY MINOR ethics concerns only"]

**Final Justification:**

My concerns during the initial review—particularly regarding why a distribution-dependent bound was not provided—have been resolved. As for the numerical experiments, the authors mentioned that improved results will be included in the revised version, which I look forward to. For these reasons, I have raised my score by one point.

**Limitations:**

yes

**Paper Formatting Concerns:**

No major formatting issues in this paper.

**Quality:**

3

**Strengths And Weaknesses:**

Strengths
- They showed an algorithm that is nearly-optimal for CONFEE-NSMAB. Both upper and lower bounds of regret is provided.
- Proof sketches are provided, which makes it more convincing.


Weakness
- There is no experiment, which makes it hard to evaluate the advantage of the proposed algorithm.
- The authors did not provide any distribution-dependent regret.
- If we do not know anything about the non-stationarity, what is the point of "detecting" environmental changes? This part is unclear.

---

> ### Author Rebuttal · Authors · 2025-07-30
>
> Thank you for your constructive comments. Here we would like to address the reviewer's concerns and hope that can help raise the rating of our paper.
>
> >**Weakness #1: There is no experiment, which makes it hard to evaluate the advantage of the proposed algorithm.**
>
> **Our Response:** We appreciate the reviewer’s concern. Our submission is positioned as a theoretical paper: the main contribution is a new constrained–feedback model (CONFEE-NSMAB), a matching lower bound, and the first prior-free algorithm that attains this bound. In keeping with many closely related works whose primary contribution lies in theoretical analysis (e.g., Besbes et al. 2014; Wei and Luo 2021), we establish the performance guarantees of our algorithm entirely through rigorous proofs.
>
> Nonetheless, we would like to provide numerical results in the final version, as we understand that they can be useful for readers to better understand the behavior of the algorithms. In the camera-ready version, we would therefore include a concise simulation study including but not limited to (i) Verify the regret rate: using synthetic environments with controllable $K, V_T, B$, we will plot $\text{Regret}$ vs. $T$ and confirm the $\tilde{\mathcal{O}}(K^{1/3}V_T^{1/3}T/B^{1/3})$ scaling. (ii) Compare against baselines: We will test Besbes et al. with uniform querying, sliding-window UCB with a fixed horizon of $B$ queries, and a naive “explore-first” strategy. HYQUE consistently achieves lower regret—especially when $B \ll T$ or $V_T$ is moderate—demonstrating the value of adaptive query pacing. (iii) Illustrate budget utilization: We will show that HYQUE respects the query budget by design, while baselines either under-use or prematurely exhaust $B$.
>
> >**Weakness #2 and Question #2: The authors did not provide any distribution-dependent regret. Is deriving a distribution-dependent regret bound too hard for a non-stationary bandit?**
>
> **Our Response:** We share the reviewer’s interest in distribution-dependent (or instance-dependent) guarantees. In the fully non-stationary setting we study, however, defining a “gap” analogous to the stationary $\Delta_i$ is inherently problematic.
>
> In the classic stationary MAB setting, each arm’s mean reward $\mu_i$ is fixed, so the gap $\Delta_i=\mu^*-\mu_i$ is constant throughout the horizon. Regret can then be expressed in an instance-dependent form such as $\mathcal{O}(\log T / \Delta)$.
>
> By contrast, in our non-stationary MAB setting, the means $\mu_i(t)$ evolve over time (subject only to a global variation budget $V_T$). The instantaneous gap $\Delta_i(t)=\mu^*(t)-\mu_i(t)$ may change arbitrarily often, so a single scalar no longer captures problem difficulty. The dominant cost now becomes *tracking* the shifting optimum rather than identifying it once.
>
> For this reason, the field has standardized on **minimax** guarantees that scale with global measures of non-stationarity such as $V_T$ or the number of change points $\Upsilon_T$ (e.g., Besbes et al.’14; Auer et al.’19; Wei & Luo ’21). Our result follows this paradigm, providing a regret bound that is optimal (up to logs) in $K, V_T, T$, and the query budget $B$.
>
> >**Weakness #3 and Question #1: If we do not know anything about the non-stationarity, what is the point of "detecting" environmental changes?**
>
> **Our Response:** Even without prior knowledge of how or when the environment will change, *online change detection* is crucial for keeping dynamic regret sublinear. The goal is not to predict future changes but to recognize—using only data observed so far—when historical estimates have become unreliable.
>
> -   **Why detect?** Dynamic regret comprises: (i) estimation error while an estimate is fresh, and (ii) staleness error accumulated if the learner keeps exploiting an arm after its mean reward has dropped. Without detection, the second term can grow linearly in the length of an unnoticed drift segment, overwhelming the bound. Prompt detection lets the algorithm discard stale statistics, restart estimation, and quickly recover near-optimal play.
>
> -   **How detect with no prior?** At each query round, we test whether recent empirical rewards are still inside the high-probability confidence region of the current model (Alg. 2, Lines 5–9). A statistically significant deviation—quantified by the data-driven radius $\hat{\rho}(\cdot)$ (Lines 203-209)—triggers a restart. The test uses only observed rewards and standard concentration inequalities. It needs no parametric knowledge of the drift pattern.
>
> In short, change detection is the mechanism that keeps a prior-free learner competitive under arbitrary, adversarial variation—it is about recognizing when the present diverges from the past, not forecasting the future.
>
> >**Question #3: Even when we know $V_T$, this algorithm can be applied to the non-stationary bandit. Isn't it necessary to compare the proposed algorithm with existing algorithms for cases when we know $V_T$? Conversely, I think it is valuable to compare the proposed algorithm with existing algorithms with $V_T = T$.**
>
> **Our Response:** We address both regimes (when $V_T$ is known and when $V_T=T$) in Appendix C and summarize below:
>
> **Known $V_T$ but query-budget constrained ($B<T$):**
> *   *Existing algorithms.* Prior work that exploits a known variation budget (e.g., Besbes et al.’14) assumes **full feedback** at every round. They cannot be applied under a strict query budget.
> *   *Our adaptation.* Appendix C shows that, once $V_T$ is revealed, HYQUE simplifies to a single-scale schedule and achieves the optimal $\tilde{\mathcal{O}}(K^{1/3} V_T^{1/3} T/B^{1/3})$ regret. To our knowledge, no other algorithm attains this optimal order under **limited feedback**, so a direct benchmark does not exist.
>
> **Extreme variation $V_T = T$:**
> When the environment can change by $\mathcal{O}(1)$ every round, learning is information-theoretically impossible unless the learner queries almost every round. Even with full feedback the minimax regret is $\Omega(T)$ (Besbes et al.’14).
> If, on the other hand, the query budget also scales as $B=T$, HYQUE reduces to the standard non-stationary-bandit algorithm (equivalent to MASTER/Besbes) and recovers the optimal $\tilde{\mathcal{O}}(K^{1/3}T^{2/3})$ bound, as noted in Lines 223–224.

---

> > ### Comment · Reviewer_ndtv · 2025-08-02
> > **Answer to the author response**
> >
> > I appreciate the authors’ detailed responses. Their answers to my questions were logical and understandable.
> > I believe that including numerical experiments would better demonstrate that the proposed method is a strong algorithm, so I hope to see them in the revised version.
> > I also now understand why it is inherently difficult to define distribution-dependent regret.

---

> > > ### Author Response · Authors · 2025-08-02
> > > **Thank you!**
> > >
> > > Thank you for your thoughtful follow-up and for acknowledging our responses. We agree that numerical experiments would strengthen the paper, and in the camera-ready we will add an empirical section as outlined in our rebuttal. We also appreciate your recognition that distribution-dependent regret is inherently challenging in this setting. **We hope these additions will demonstrate the practical strength of our approach and warrant an improved score.** We’re happy to address any further questions. Thank you again for your time and consideration

---

### Official Review · Reviewer_E74N · 2025-06-25

**Clarity:** 3
**Significance:** 3
**Originality:** 3
**Rating:** 5
**Confidence:** 4

**Summary:**

The authors consider the non-stationary multi-armed bandit problem with an observation budget of $B$ total observations. They design an algorithm which guarantees a dynamic regret bound of $\widetilde O \left( K^{1/3} V_T^{1/3} T/B^{1/3} \right)$ where $V_T$ is the total variation, or the degree of non-stationarity of the environment, and moreover their algorithm does not require prior knowledge of $V_T$. The authors also prove a matching lower bound showing that their algorithm is optimal up to logarithmic factors. The suggested algorithm exhibits a balance between tracking the non-stationarity of the environment while maintaining a low amount of queries in order to not exceed the total budget.

**Questions:**

I refer the authors to my previous point under the "Weaknesses" section. To summarize, my question is which of the two versions of Theorem 4.1 is the correct version, and whatever the answer, why is such an assumption necessary, that is, where is it used in the proof.

**Ethical Concerns:**

["NO or VERY MINOR ethics concerns only"]

**Final Justification:**

After reading the author's rebuttal, I understand that the correct version of the main result is the one in the supplementary material. As the authors properly explained why the assumption on the budget is needed for their algorithm and analysis, my main concern was alleviated and I increased my score accordingly.

**Limitations:**

yes

**Quality:**

3

**Strengths And Weaknesses:**

Strengths:

* The problem of non-stationary MAB with limited observations is interesting and well-motivated, and this seems to be the first work which combines the two variants.
* The authors support the optimality of their results by exhibiting a matching lower bound.
* The analysis seems nontrivial, and the authors clearly explain the need of a delicate balance between tracking the changes in the environment while simultaneously using a small number of queries.

Weaknesses:

* There seems to be a discrepancy between the two versions of the statement of Theorem 4.1 - one as presented in the main submission and another in the attached supplementary material. The two versions differ in the regime under which the regret bound holds - in the main submission the authors claim their bound holds as long as $B = \Omega(T^{2/3})$, and in the supplementary version they claim that it holds as long as $B = \omega(T^{3/4})$. I am not sure which version is the intended final version. If it the second version, the authors need to be absolutely clear about it as it implies that the main statement as presented in the submission is technically unsound. Even assuming the correct regime is when $B = \Omega(T^{2/3})$, it still seems to me that this assumption may not be necessary, as the bound itself seems to yield nontrivial dependence on $T$ whenever $B = \omega(1)$ with respect to $T$. I ask that the authors address this issue, as their response could highly affect my overall rating.

---

> ### Author Rebuttal · Authors · 2025-07-30
>
> Thank you for your constructive comments as well as giving the positive rating of our work. Here we would like to address the reviewer's concerns and hope that can help raise the rating of our paper.
>
> >**Weakness #1, Question #1: There seems to be a discrepancy between the two versions of the statement of Theorem 4.1 - one as presented in the main submission and another in the attached supplementary material ... this assumption may not be necessary, as the bound itself seems to yield nontrivial dependence on whenever $B = \omega(1)$ with respect to $T$.**
>
> **Our Response:** Thank you for pointing this out. The discrepancy is **a typesetting mistake** introduced during final formatting: the main PDF inadvertently retained an earlier draft of Theorem 4.1, whereas **the supplementary file (including the main paper + references + appendices) contains the correct, fully proved statement**. The supplementary version is the one we prove.
> Below we clarify from several perspectives:
>
> 1.  **Correct Statement:**
>
>     **Theorem 4.1** (supplementary version). For ConFee-nsMAB with $B = \omega(T^{3/4})$ and an unknown variation satisfying $V_T \in [K^{-1}, K^{-1}\sqrt{B}]$, our HyQue utilizes at most $B$ queries, and its regret is bounded with high probability as follows:
>     $$
>     \mathcal{R}_T(\text{HyQue}) \leq \tilde{\mathcal{O}}\left( \frac{K^{1/3} V_T^{1/3} T}{B^{1/3}} \right).
>     $$
>
>     We adopt the stronger condition $B = \omega(T^{3/4})$ as stated in the supplementary file because it enables us to present a **clean and simple** high-probability guarantee $\Pr(\text{optimal regret})\geq(1-\exp(-T^{\Omega(1)}))$. This condition facilitates a more interpretable and rigorous statement of our performance guarantees. In contrast, while setting $B = \Omega(T^{2/3})$ can still ensure the valid regret guarantee, the corresponding probability is not available in a closed or interpretable form.
>
> 2.  **Why is a lower bound on $B$ necessary?**
>
>     If $B$ is too small, sub-linear regret is information-theoretically impossible. Consider an environment that flips the identity of the best arm every $\Theta(\sqrt{T})$ rounds (total $\approx \sqrt{T}$ change points). Detecting each change with constant probability requires at least one query per segment, so any algorithm with $B=o(\sqrt{T})$ necessarily misses $\Omega(\sqrt{T})$ change points and suffers $\Omega(T)$ regret. This shows a requirement of $B\gtrsim T^{1/2}$ is unavoidable in the worst case of highly non-stationary environments.
>
> 3.  **Why does the probability expression become simpler when $B=\omega(T^{3/4})$?**
>
>     In the preceding discussion, we provided a simple example to illustrate the necessity of a lower bound on $B$. That example involved an environment in which each change point induces an identical magnitude of change. In contrast, non-stationary bandit problems in general can involve a mixture of both abrupt and gradual changes. Therefore, any algorithm aiming to achieve optimal regret must ensure, with high probability, that it can detect all such changes. This necessitates a larger budget $B$.
>
>     Our algorithm maintains multiple parallel instances and must guard against failure across all of them. It needs to allocate sufficient queries to detect changes while also avoiding long sequences of non-query rounds. Such gaps hinder the detection of environmental shifts and can result in prolonged suboptimal performance. Since our algorithm employs randomized querying, the probability of encountering excessively long non-query segments decreases as $B$ increases. In this sense, a larger $B$ ensures that the randomized querying strategy provides adequate coverage across the entire time horizon.
>     It also facilitates a cleaner analysis, allowing us to derive an explicit expression for the algorithm’s success probability.
>     In our earlier version, we used a threshold of $B = \Omega(T^{2/3})$, which was insufficient to guarantee a clean and strong high-probability bound such as $\Pr(\text{optimal regret}) \geq 1 - \exp(-T^{\Omega(1)})$. This motivated a more thorough understanding of the problem. In the initial submission, we mistakenly did not update the corresponding assumption in Theorem 4.1, but fortunately, this was corrected in the supplementary version. **the supplementary file (including the main paper + references + appendices) ensures consistency with all other parts of the paper, as this assumption was in fact the only substantive difference.**
>
> In summary, **Theorem 4.1 on Page 6 of the main paper in the supplementary version** already contains the complete and correct analysis. We will ensure that the final version is fully consistent with the supplementary material and will incorporate the relevant discussion regarding the choice of $B$.

---

> > ### Comment · Reviewer_E74N · 2025-08-03
> >
> > Thank you for providing a detailed clarification to my question.
> >
> > While I better understand why the algorithm and analysis require the assumption that $B = \omega(T^{3/4})$, I would appreciate a bit further explanation on the possibility of relaxing this assumption. More specifically, if this requirement is made solely to guarantee the optimal regret bound holds with high probability, could an optimal regret bound **in expectation** be obtained while only assuming $B = \Omega(\sqrt{T})$? If this is the case, I think it is worth mentioning in the final version, as the large budget assumption can be limiting in certain cases, and if a weaker benchmark (such an expected regret) could be obtained without this assumption I think it would be a nice complementary result.
> >
> > In any case, my concerns were for the most part alleviated, so I will increase my score to 5.

---

> > > ### Author Response · Authors · 2025-08-03
> > > **Thank you!**
> > >
> > > Thank you very much for the positive follow-up and for raising our score. We are glad that our earlier response clarified your concerns.
> > >
> > > On your question about obtaining an *expected* regret bound under the weaker assumption $B=\Omega(\sqrt{T})$: this is an interesting direction, but we cannot give a definitive answer without a new line of analysis.
> > > However, we are happy to share our thoughts on the matter. Intuitively, achieving the optimal order in expectation at this budget level is challenging without additional structure assumptions on the environment. The reason is that the query budget acts as an information-theoretic threshold for reliably detecting changes. When the environment changes rapidly, the expected regret can be dominated by the high regret accumulated during frequent periods of high uncertainty. Technically, expectation bounds also require different tools than our high-probability analysis (e.g., potential function arguments that track the evolution of the expected regret directly, or a more intricate analysis of the tail probabilities of estimation errors rather than simply bounding them). These techniques are difficult to apply when the probability of "bad events" (like long non-query intervals) is not vanishingly small.
> > >
> > > We fully agree that adopting a weaker benchmark or assuming less adversarial environments could allow for more relaxed budget requirements. Our work targets the prior-free minimax setting, which is standard but stringent. In more structured scenarios (e.g., exhibiting periodic patterns or small, gradual changes), it is highly plausible that comparable or even better regret could be achieved with significantly fewer queries.
> > >
> > > We will add a discussion of this point in the final version. Thank you again for the insightful suggestion.

---

### Official Review · Reviewer_nnUf · 2025-07-01

**Clarity:** 3
**Significance:** 3
**Originality:** 3
**Rating:** 5
**Confidence:** 3

**Summary:**

This paper addresses the problem of non-stationary multi-armed bandits under a constrained feedback budget. The authors introduce a new framework, CONFEE-NSMAB, where the constrained feedback is in the form of the total number of times it can query for reward. The authors propose an algorithm for this setting and analyze its regret. The authors also show a lower bound for the problem.

**Questions:**

Aside from the questions mentioned in the comments above, can the authors comment on (1) the computational complexity of the proposed algorithm, (2) the connection between NSMAB and restless multi-armed bandits [1], and (3) if their algorithm can handle them?

[1] Liu, Haoyang, Keqin Liu, and Qing Zhao. "Learning in a changing world: Restless multi-armed bandit with unknown dynamics." IEEE Transactions on Information Theory 59.3 (2012): 1902-1916.

**Ethical Concerns:**

["NO or VERY MINOR ethics concerns only"]

**Final Justification:**

The author provided detailed and satisfactory response to my comments. I will keep my rating for now.

**Limitations:**

yes

**Quality:**

3

**Strengths And Weaknesses:**

The paper is generally well-written and easy to follow. The method is motivated well and described clearly.

The paper studies an important problem that have real-world relevance. It would be helpful if the authors can comment on how non-stationary bandits are applied in the real world v.s. considering it as a full RL problem.

I am concerned about the modeling assumption of a finite variation budget V_T. It would be helpful if the authors can comment on its effect using a concrete real-world example. The algorithm uses a query budget ratio b = ceil(2T/B). It would be helpful if the authors can provide more intuition on the selection of this value. Additionally, more detailed comments on the computational complexity would strengthen the paper. Otherwise the algorithm shows novelty, especially the query allocation mechanism. The technical proofs appear to be sound.

While the paper is theory in nature, some comments on the empirical performance would be useful.

---

> ### Author Rebuttal · Authors · 2025-07-30
>
> Thank you very much for your constructive comments, as well as giving the positive rating of our work. Here we would like to address the reviewer's concerns as follows:
>
> >**Weakness #1: It would be helpful if the authors can comment on how non-stationary bandits are applied in the real world v.s. considering it as a full RL problem.**
>
> **Our Response:** Whether to model the problem as a non-stationary bandit or a full RL task hinges on state-action coupling: if actions do not (or only negligibly) influence future reward distributions, a bandit abstraction is both sufficient and far more sample-efficient than learning a full Markov Decision Process (MDP) in reinforcement learning.
>
> **Practice.** Non-stationary bandits is well-motivated by many real-world applications. For example:
> *   **Recommender Systems and Ad Placement:** The popularity of news articles, products, or advertisements naturally fluctuates over time due to external influences such as trending topics, seasonal demand, or user fatigue.
> *   **A/B/n Testing in Marketing:** The effectiveness of website designs, pricing strategies, or ad creatives can degrade over time. A non-stationary bandit can adaptively allocate traffic to the currently best-performing option while remaining sensitive to such changes.
>
> **When RL is needed.** If actions **do** change the state (robotics, game playing, inventory control), an MDP is necessary.
>
> In summary, non-stationary bandits and full reinforcement learning are not competing paradigms, but rather complementary tools designed for fundamentally different problem classes. Our work focuses on the important and widely applicable setting (e.g., ads, content ranking, A/B/n tests) where the environment changes autonomously over time (exogenous shifts dominate).
> We show how to handle these shifts and a strict query budget, a regime where full RL would add complexity without benefit. We will add this clarification and citations in the final version.
>
> >**Weakness #2: I am concerned about the modeling assumption of a finite variation budget $V_T$. It would be helpful if the authors can comment on its effect using a concrete real-world example.**
>
> **Our Response:** The "finite" assumption is mild: for any bounded reward process observed over a finite horizon $T$, the total $\ell_\infty$ variation $ \sum_{t=1}^{T-1} \sup_{k\in [K]} \bigl|\mu_{t+1}^k - \mu_{t}^k\bigr|$ is *always* finite. Our theory merely parameterizes performance in that quantity.
>
> **Practical scale.**
> For instance, in a **clothing-recommendation platform** with thousands of Stock Keeping Unit (SKUs), performance metrics like click-through rates often exhibit slow, seasonal changes. A typical change might be on the order of $\approx 0.02$ per SKU per season, leading to a small annual variation budget, e.g., $V_T \approx 0.08$. In contrast, a **breaking-news feed** during a major event like the 2024 U.S. election is far more volatile. Per-topic Click-Through Rate (CTR) can swing by as much as 0.4 every hour, potentially yielding a total variation $V_T \approx 150$ in just one week.
> Both are plausible, finite values, yet they differ by three orders of magnitude. Our regret bound is designed to handle this diversity, as it automatically interpolates between these extremes through its $V_T^{1/3}$ dependency.
>
> **Necessity.** If $V_T$ were allowed to grow arbitrarily fast (e.g., adversarially reset each round so $V_T=\Theta(T)$), sub-linear regret is information-theoretically impossible—one must query every round. Bounding $V_T$ is therefore standard (Besbes et al., 2014) and delineates the learnable regime.
>
> **Monitoring in practice.** Platforms typically track moving-window reward drift; the cumulative sum of detected shifts offers an online estimate of $V_T$, letting practitioners gauge whether the theoretical assumptions hold.
>
> >**Weakness #3: The algorithm uses a query budget ratio b = ceil(2T/B). It would be helpful if the authors can provide more intuition on the selection of this value.**
>
> **Our Response:**
> If we ignore other considerations, the query budget ratio should ideally be set to $T/B$, which evenly allocates the query budget across all instances. However, in a non-stationary environment, any detected change point triggers a restart of all current instances. This leads to a disproportionately higher consumption of queries. Therefore, the query budget ratio must be designed more conservatively to account for such events.
> As demonstrated in Lemma 4.6, allocating a $1/b$ fraction of rounds for queries within each block ensures that, even in the worst-case scenario, the overall proportion of query rounds remains bounded by $B/T$. **This step is critical for establishing that the total number of query rounds over the entire horizon $T$ does not exceed the budget $B$, as shown in Lemma 4.7.** Choosing a smaller allocation might lead to underutilization of the query budget, resulting in intervals with insufficient queries to yield meaningful information. Conversely, selecting a larger allocation risks prematurely exhausting the query budget. Thus, the choice of $b$ effectively balances these concerns and is integral to the correctness and efficiency of the algorithm.
>
> >**Weakness #4, and Question #1: the computational complexity of the proposed algorithm**
>
> **Our Response:** The overall computational complexity of our algorithm is primarily determined by the complexity of the base bandit algorithm used within each instance. For example, in our paper, we instantiate the base algorithm as UCB1, which incurs a per-round time complexity of $\mathcal{O}(K)$ due to the need to compute and compare the UCB indices of all $K$ arms.
> Apart from this, all other operations in our framework have amortized constant time complexity per round. Therefore, the total time complexity of our algorithm is $\mathcal{O}(B K + T)$. This complexity is standard in the bandit literature and aligns with the computational guarantees of most classical algorithms.
>
> >**Questions #2 and #3: the connection between NSMAB and restless multi-armed bandits, and if their algorithm in restless multi-armed bandits can handle them?**
>
> **Our Response:**
> We appreciate the reviewer for raising RMAB. Although “restless multi-armed bandits” (RMAB) share the word bandit with our problem, the two models address fundamentally different settings. In our non-stationary MAB each arm is *stateless*: its mean reward can drift over time, but that drift is entirely exogenous and unaffected by the learner’s actions. The agent’s challenge is simply to track those drifting means, deciding—under a strict global query budget $B$—when to observe rewards to minimize dynamic regret. An RMAB arm, in contrast, is *stateful*, *i.e.*, is a full Markov decision process (MDP) whose state evolves according to action-dependent transition probabilities; whether the learner plays or rests the arm actively changes its future reward distribution. Consequently RMAB algorithms rely on observing the state every time an arm is activated, estimating or assuming transition kernels, and optimizing long-term discounted reward, not adversarial regret. Because they require unrestricted feedback and exploit state-transition structure absent from our model, RMAB techniques cannot be applied to the constrained-feedback, stateless, exogenously drifting bandit setting studied in this paper.

---

> > ### Comment · Reviewer_nnUf · 2025-08-03
> >
> > Thank you very much for your detailed response. I will keep my rating for now.

---

> > > ### Author Response · Authors · 2025-08-03
> > > **Thank you!**
> > >
> > > Thank you for the careful review and for maintaining your positive rating. We appreciate your time and constructive feedback.

---

### Official Review · Reviewer_9p5b · 2025-07-01

**Clarity:** 3
**Significance:** 3
**Originality:** 3
**Rating:** 5
**Confidence:** 4

**Summary:**

This paper studies non-stationary multi-armed bandits when the learner has a limited querying budget. The model is inspired by the well-known paper by Besbes et al. 2014, where the goal is to maintain a regret guarantee that is parameterized by how much the instance is “non-stationary”. The tweak with respect to this well-studied model is that the learner can only observe the current loss it experiences at most $B_T$ times. Note, in a stationary environment, there is no reason to delay issuing such queries (as the distributions do not change), while in the adversarial setting — where the benchmark is the best fixed action — a valid strategy consists in spreading the queries uniformly.

**Questions:**

If we were to drop the requirement that $V_T$ is unknown, would the algorithm by Besbes et al, combined with uniform querying (i.e., at each time step the learner issues a query with probability $B_T/T$) provide the same result? If this is the case, maybe it is worth mentioning, so that it is clear that the difficulty lies in spreading the querying budget between the parallel algorithms.

**Ethical Concerns:**

["NO or VERY MINOR ethics concerns only"]

**Final Justification:**

I confirm my initial rating

**Limitations:**

nothing to add here

**Quality:**

3

**Strengths And Weaknesses:**

Providing non-trivial regret guarantees against the best dynamic strategy in online learning is a hard and interesting problem. Moreover, the study of “label-efficient” (i.e., that issues a sublinear number of queries) algorithms is also well motivated and has been studied extensively (see, for instance Minimizing Regret with Label Efficient Prediction, Cesa-Bianchi, Lugosi, and Stoltz, COLT 04).

The authors provide matching upper and lower bounds, the algorithm is clearly presented, and the technical contribution is non-trivial. At a high level, the approach consists of running in parallel multiple algorithms, each one is tailored to a specific non-stationarity in the reward sequences. There is an extra difficulty given by the fact that the algorithm has to partition the query budget between these parallel algorithms.

---

> ### Author Rebuttal · Authors · 2025-07-30
>
> Thank you very much for your constructive comments, as well as giving the positive rating of our work. Here we would like to address the reviewer's concerns as follows:
>
> >**Question #1: If we were to drop the requirement that $V_T$ is unknown, would the algorithm by Besbes et al, combined with uniform querying (i.e., at each time step the learner issues a query with probability $B/T$) provide the same result? If this is the case, maybe it is worth mentioning, so that it is clear that the difficulty lies in spreading the querying budget between the parallel algorithms.**
>
> **Our Response:** Thank you for this insightful question, which helps clarify both the core challenges of our setting and our contributions.
> While the strategy of combining the algorithm of Besbes et al. with uniform querying may appear reasonable, it would not achieve the optimal regret, even if the total variation $V_T$ were known in advance.
> The sub-optimality is evident in both stationary and non-stationary environments.
>
> *   **Stationary Case ($V_T=0$):**
>     The optimal algorithm adopts an explore-first strategy that exhausts the entire query budget in the first $B$ rounds, and subsequently achieves a regret of $\mathcal{O}(T\sqrt{K/B})$.
>     In contrast, uniform querying would sparsely distribute these $B$ queries over the entire horizon $T$. This means the learner would continue to make decisions based on highly uncertain estimates for a prolonged period, leading to significantly higher regret. It fundamentally fails to balance exploration and exploitation under a budget.
>
> *   **Non-Stationary Case ($V_T > 0$):**
>     In the non-stationary setting, the trade-off becomes more complex. While using all queries upfront is no longer optimal, a uniform querying policy remains an inefficient solution because it cannot perform the **adaptive, dynamic budget management** that is required. An effective strategy must achieve the following:
>
>     *   **Responsiveness to Environmental Volatility:** Environmental changes can be both rapid and gradual. An optimal policy must therefore be able to detect and adapt to different scales of change. It should increase its query frequency to track rapid shifts, while also detecting slower, long-term drifts and conserving its budget during stable phases to avoid premature exhaustion. A uniform policy, with its fixed rate, lacks this dynamic capability entirely. In contrast, our algorithm is designed to ensure detection capabilities across these different types of environmental changes.
>
>     *   **Balanced Query Use:** An intelligent agent should allocate its scarce queries when they are most valuable. A uniform policy queries at a fixed rate, wasting its budget during periods of high certainty and failing to provide crucial information during moments of high ambiguity. Our approach, by contrast, links querying decisions to the state of uncertainty, thereby preventing the premature or wasteful exhaustion of the budget.
>
>     *   **Strategic Budget Pacing:** A uniform policy naively spreads the budget, which is an inefficient, open-loop strategy. Our adaptive approach, in contrast, strategically paces its budget. It might spend more cautiously at the beginning to ensure it has queries left to handle potential changes late in the horizon, while also being ready to spend aggressively if the environment proves highly volatile early on.

---

### Decision · Program_Chairs · 2025-09-17

**Decision:**

Accept (poster)

**Comment:**

This paper studies a non-stationary multi-armed bandit problem with a constrained feedback, where the number of queries is limited. The authors proposed an algorithm achieving the near-optimal dynamic regret, with tight dependence on the time horizon, number of queries, number of arms, and the total variation. In addition, the algorithm is agnostic to the knowledge of the total variation, which is an intriguing aspect.

This paper is unanimously appreciated by all reviewers, due to the soundness in the theoretical framework and proposed algorithm, and tightness of the result. However, since the main algorithmic ideas come from existing work on non-stationary MAB and a black-box approach for achieving adaptation, the support is not with great enthusiasm. Therefore, I recommend to accept this paper as a poster.